# CoGeoAD: Hierarchical Color-Geometric Fusion with Multi-View Attention for Zero-Shot 3D Anomaly Detection

Ke Xu [1 2]   Xinle Wang [2]   Yanning Hou [3]   Xueliang Ma [2]   Juan Xie [4]   Jianfeng Qiu* [2]

## Abstract

Zero-shot 3D anomaly detection is essential for industrial quality inspection, where labeled anomaly samples are scarce. Meanwhile, existing methods lack an effective mechanism to fuse complementary 2D color images with 3D geometric structures, limiting their ability to detect both surface and structural defects in a unified framework. To address these issues, we propose CoGeoAD, a unified CLIP-based framework that fuses color and geometric features by constructing pixel-aligned paired multi-view images. The framework introduces a Data-Driven Multi-View Attention (MVA) mechanism to adaptively aggregate 3D features and a Multi-Stage Color-Geometric Fusion (MS-CGF) module to hierarchically integrate multilevel features from both modalities. Extensive experiments on the MVTec3D-AD and Eyecandies benchmarks demonstrate that CoGeoAD achieves state-of-the-art performance, effectively capturing both structural and textural anomalies in complex industrial scenarios.

## 1. Introduction

The surge in intelligent manufacturing and industrial automation has catalyzed a critical need for anomaly detection (AD) (Bergmann et al., 2020; Tao et al., 2022a; Gu et al., 2024). However, in real-world industrial settings, obtaining a large amount of annotated anomaly data is often prohibitively difficult (Tao et al., 2022a;b). Anomalies are inherently rare, leading to highly imbalanced data distributions (Bergmann et al., 2019; 2021; Bonfiglioli et al., 2022;

---

[1]State Key Laboratory of Opto-Electronic Information Acquisition and Protection Technology, Anhui University, Hefei, China [2]School of Artificial Intelligence, Anhui University, Hefei, China [3]College of Intelligence Science and Technology, National University of Defense Technology, Changsha, China [4]School of Mathematics & Physics, Anhui Jianzhu University, Hefei, China. Correspondence to: Jianfeng Qiu <qiujianf@ahu.edu.cn>.

*Proceedings of the 43rd International Conference on Machine Learning*, Seoul, South Korea. PMLR 306, 2026. Copyright 2026 by the author(s).

Wang et al., 2024; Gao, 2024; Huang et al., 2025; Dai et al., 2025). Moreover, data acquisition is costly and often constrained by privacy and safety concerns (Zhou et al., 2023). Given these data constraints, zero-shot anomaly detection, which eliminates the dependence on target-domain training data, has emerged as a critical research direction. More broadly, recent studies have explored soft reasoning and transferable representation learning to improve generalization in data-scarce settings (Hou et al., 2025c). In visual anomaly detection, the community has turned to foundational vision-language models (VLMs), particularly CLIP model (Radford et al., 2021). With its remarkable cross-modal semantic understanding and generalization capabilities, CLIP offers a powerful foundation for addressing the challenges of zero-shot anomaly detection.

While conventional AD methods have flourished in 2D vision (Huang et al., 2022; You et al., 2022; Cao et al., 2024b), their exclusive reliance on RGB data inherently neglects crucial geometric information. Conversely, emerging 3D point cloud-based approaches (Bergmann & Sattlegger, 2023; Chen et al., 2023a; Chu et al., 2023) excel at capturing spatial details but often fail to detect defects manifested as color or material texture (see Fig. 1(a)). Recently, CLIP-based image anomaly detection tasks have achieved remarkable success (Jeong et al., 2023; Chen et al., 2023b; Cao et al., 2024c; Qu et al., 2024; Gao et al., 2025), but its potential in the 3D domain still needs to be explored. Although some methods (Cheng et al., 2025b; Zhou et al., 2024a; 2025) attempt to utilize 2D-RGB image information in 3D point cloud settings, they typically treat RGB images merely as auxiliary cues without integrating color information into the geometry. Since texture anomalies are affected by viewpoint, the aforementioned methods cannot effectively achieve anomaly perception of 2D color across multiple 3D viewpoints. Consequently, establishing an effective synergy between color and geometric modalities remains a pivotal challenge.

To bridge this gap, we propose CoGeoAD, a unified CLIP-based framework that fuses color and geometric features. Distinct from simple ensembling, our framework leverages paired geometric and color renderings with explicit point-to-pixel correspondences. This allows us to isolate structural

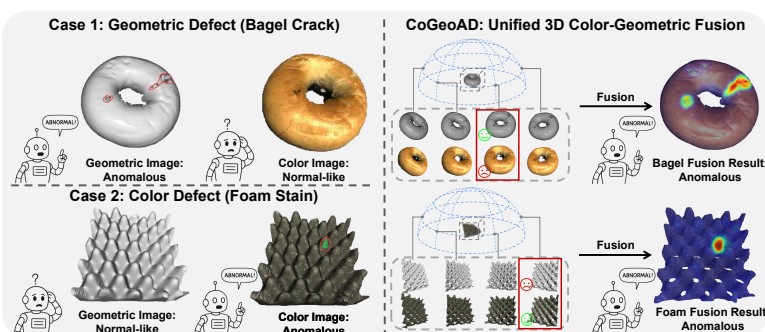

*(a)* Visual complementarity of modalities.

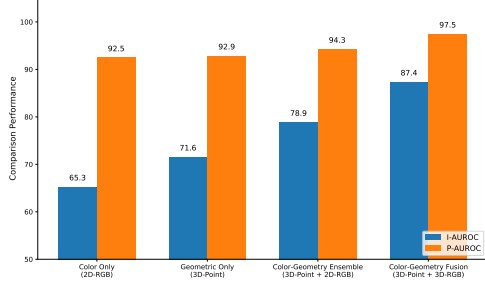

*(b)* Performance comparison with baselines.

*Figure 1.* **Motivation and Performance Analysis. (a) Visual Complementarity:** The left panel illustrates the limitations of single modalities: geometric features miss color defects (e.g., Foam stain), while RGB features overlook structural cracks (e.g., Bagel crack). In contrast, the right panel shows our CoGeoAD framework, which fuses multi-view geometric and color representations to accurately detect and localize both types of anomalies. **(b) Performance comparison:** Quantitative comparison on MVTec3D-AD dataset demonstrates that our proposed fusion method significantly outperforms single-modal baselines and simple ensemble.

anomalies via geometric projections while simultaneously capturing texture defects through color views. By fusing modalities outside the frozen CLIP encoder, our approach preserves the model's generalizability and avoids the internal network modifications that typically cause catastrophic forgetting.

Furthermore, to improve feature aggregation without inheriting biases from pretrained models, we introduce a Data-Driven Multi-View Attention (MVA) mechanism derived directly from multi-view images. Finally, to robustly synthesize these representations, we propose the Multi-Stage Color-Geometric Fusion (MS-CGF) module. This module hierarchically integrates multi-level features from both modalities, ensuring the detection of subtle anomalies that might be missed by single-modal approaches. Fig. 1(b) compares the progressive performance improvements of the methods from 2D-RGB (color only), 3D-Point (geometry only), 2D-RGB+3D-Point (color-geometry ensemble) to the proposed 3D-Point+3D-RGB (color-geometry fusion) on the MVTec3D-AD dataset. Our key contributions are as follows:

- We propose CoGeoAD, a zero-shot framework that uses dual-modal rendering to generate paired geometric and color views. These views are processed by CLIP and mapped back to 3D space using point-to-pixel correspondence to eliminate occlusion noise.

- We introduce a model-agnostic MVA mechanism that computes attention weights directly from images rather than intermediate feature embeddings. By avoiding dependence on pretrained model biases, MVA significantly enhances the robustness of 3D feature aggregation for identifying subtle industrial anomalies.

- We propose the MS-CGF module to hierarchically aggregate multi-layer CLIP features. It employs dynamic layer selector and modality-specific maximization to

effectively capture multi-level anomalies across color and geometry, ensuring complete detection ranging from surface defects to structural irregularities.

- Extensive experiments on the MVTec3D-AD and Eyecandies benchmarks demonstrate that CoGeoAD outperforms existing zero-shot approaches, establishing a new state-of-the-art for complex industrial anomaly detection tasks. For reproducibility, our source code is available at https://github.com/kingdomShu/CoGeoAD.

## 2. Related work

### 2.1. Unsupervised Point Cloud Anomaly Detection

Unsupervised 3D anomaly detection predominantly relies on reconstruction, memory banks, and multimodal feature learning. Reconstruction-based methods detect anomalies through high reconstruction errors of abnormal samples (e.g., IMRNet (Li et al., 2024), R3D-AD (Zhou et al., 2024b)). Recent approaches like PASDF (Zheng et al., 2025), DUS-Net (Liang et al., 2025b), MC3D-AD (Cheng et al., 2025a), and Simple3D (Cheng et al., 2026) further advance this paradigm by improving geometric modeling fidelity and targeting high-resolution industrial inspection. Memory-bank methods identify defects by comparing test features against stored normal prototypes (e.g., Reg3D-AD (Liu et al., 2023)), while frameworks such as M3DM (Wang et al., 2023), CPMF (Cao et al., 2024a), ISMP (Liang et al., 2025a), and BridgeNet (Xiang et al., 2025) incorporate multimodal cues and spatial views to construct robust memory representations. Despite significant progress in reconstruction quality and multimodal fusion, most of these methods still require category-specific or dataset-level training, inherently limiting their flexibility for zero-shot deployment.

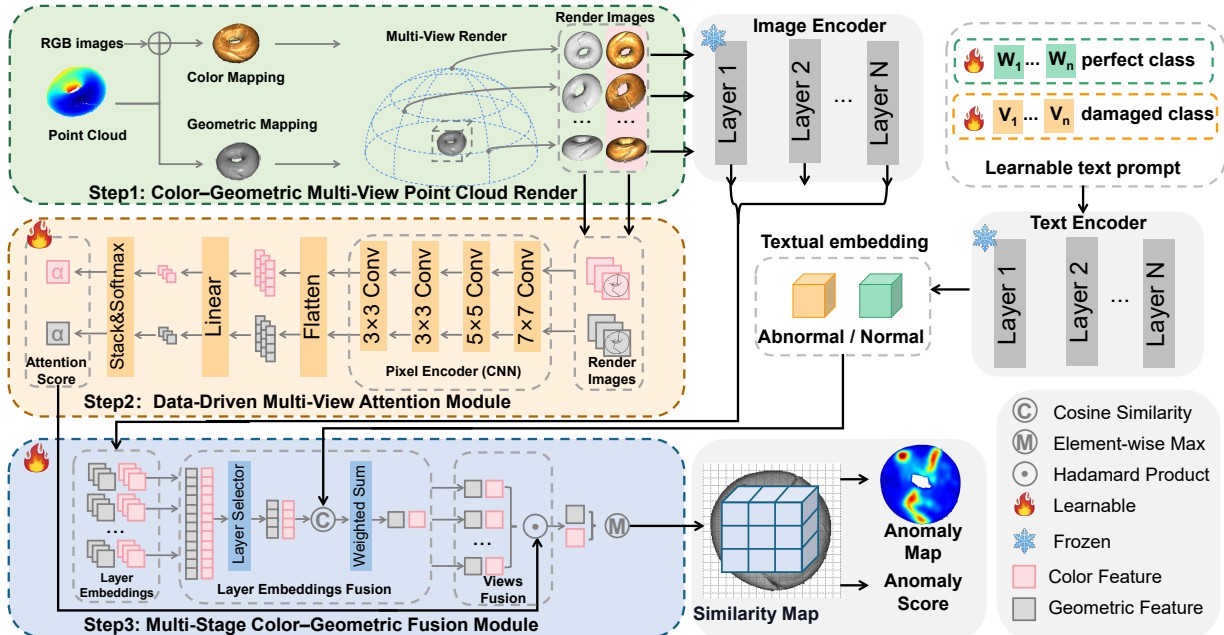

*Figure 2.* Framework of CoGeoAD. The pipeline starts by aligning RGB images with point clouds to create paired color and geometric multi-view images (Step1). A CNN-based pixel encoder assigns importance scores to each view and adaptively merges them (Step2). Multi-layer features from CLIP image encoder are selected, matched with text embeddings, and fused via learnable weights (Step3). Finally, color and geometric anomaly maps are combined through element-wise maximum to generate the final anomaly map and score.

## 2.2. Zero-shot Image Anomaly Detection

Zero-shot image industrial anomaly detection has rapidly advanced, primarily driven by CLIP. WinCLIP (Jeong et al., 2023) pioneered this field by establishing the baseline, utilizing a sliding window and multi-level visual feature fusion to capture fine-grained details. To enhance patch-level localization, APRIL-GAN (Chen et al., 2023b) introduced a lightweight linear projection layer for visual feature re-encoding, though it left the text-space ambiguity unresolved. Addressing the text side, AnomalyCLIP (Zhou et al., 2023) first adopted learnable text prompts to dynamically enhance anomaly semantics. Building on this, AdaCLIP (Cao et al., 2024c) proposed a static and dynamic learnable prompting strategy. VCP-CLIP (Qu et al., 2024) introduces a Visual Context Prompting framework that injects learnable image-level visual context into the CLIP architecture. Recent studies further improve zero-shot industrial anomaly detection by exploring cascaded CLIP-SAM prompting, clustering-driven stacked prompts, and language-free ViT-based anomaly modeling (Hou et al., 2025b;a; 2026).

## 2.3. Zero-shot Point Cloud Anomaly Detection

Recently, the scarcity of 3D data has driven research toward CLIP-based and training-free zero-shot methods. Point-CLIP (Zhang et al., 2022) and PointCLIP V2 (Zhu et al., 2023) pioneered projecting 3D data into multi-view 2D images for classification. Adapting this for anomaly detection, MVP-PCLIP (Cheng et al., 2025b) utilized depth images

with specialized visual and text prompts, while PointAD (Zhou et al., 2024a) enhanced fine-grained localization by employing high-precision rendering and view masks. Subsequently, PointAD+ (Zhou et al., 2025) moved beyond pure projection by introducing explicit 3D representation modeling to capture geometric relationships directly. More recently, MuSc-V2 (Li et al., 2025) explored zero-shot multimodal industrial anomaly classification and segmentation by mutually scoring unlabeled 2D and 3D samples without additional fine-tuning. GS-CLIP (Deng et al., 2026) further introduces geometry-aware prompts and synergistic view representation learning to adapt CLIP for zero-shot 3D anomaly detection. Compared with these concurrent and recent methods, CoGeoAD focuses on hierarchical color-geometric fusion with data-driven multi-view attention, explicitly exploiting paired chromatic and geometric renderings while preserving the generalization ability of the frozen CLIP encoder.

## 3. Methodology

### 3.1. Overview

As illustrated in Fig. 2, CoGeoAD is a zero-shot framework designed to bridge the domain gap by synergizing 2D semantic priors with 3D geometric representations. The pipeline comprises three integral modules: (1) Color-Geometric Multi-View Point Cloud Render, which establishes pixel-wise correspondences between RGB images and point clouds to generate chromatic and geometric views. (2)

Data-Driven Multi-View Attention (MVA), which utilizes a lightweight CNN-based Pixel Encoder to dynamically compute attention scores from multi-view images, ensuring that informative viewpoints are prioritized; and (3) Multi-Stage CoGeo Fusion (MS-CGF), which hierarchically aggregates multi-level features and computes text-image similarities to generate anomaly maps.

### 3.2. Color-Geometric Multi-View Point Cloud Render

To transfer 2D anomaly detection to the 3D domain, the Color-Geometric Multi-View Point Cloud Rendering module acts as the foundational preprocessing stage. It renders point clouds into paired color–geometry images under multiple views, establishing explicit point-to-pixel correspondences for structured 2D representations.

To capture surface details of complex 3D objects, we generate multi-view representations by rotating the input point cloud around its geometric center to reduce self-occlusion. The view set is defined on a discrete grid of rotation angles $\theta_x \in \Theta_x$ and $\theta_y \in \Theta_y$, where each view is obtained via the corresponding composite rotation matrix:

$$\mathbf{R}_{ij} = \mathbf{R}_y(\theta_j)\mathbf{R}_x(\theta_i), \quad \forall \theta_i \in \Theta_x, \theta_j \in \Theta_y. \quad (1)$$

CoGeoAD decouples geometry from color via dual-modal rendering. For each viewpoint, we generate geometric multi-view images by rendering the point cloud with a uniform neutral grey color against a white background, effectively isolating structural anomalies from color interference. Concurrently, color multi-view images are rendered using the chromatic information to capture color-dependent defects like stains.

To enable pixel-precise anomaly localization back onto the 3D object, we compute a dense correspondence map for every generated view. For a rotated 3D point $\mathbf{p}'_k = (x', y', z')^T$, its projection to the 2D image plane coordinate $\mathbf{x}_k = (u, v)^T$ is calculated using the pinhole camera model:

$$s \begin{bmatrix} u \\ v \\ 1 \end{bmatrix} = \mathbf{K} \left[ \mathbf{R}_{cam} \mid \mathbf{t}_{cam} \right] \begin{bmatrix} x' \\ y' \\ z' \\ 1 \end{bmatrix}, \quad (2)$$

where $s$ denotes the depth scale factor in camera coordinates, $\mathbf{K}$ is the intrinsic matrix and $[\mathbf{R}_{cam} \mid \mathbf{t}_{cam}]$ is the extrinsic matrix determined by the virtual camera's pose settings.

Crucially, we address the occlusion problem by calculating a visibility mask. Since multiple 3D points may project to the same 2D pixel coordinate $\pi(\mathbf{p})$, we enforce a geometric consistency check based on the camera-space depth $z'$. The visibility indicator $M_k \in \{0, 1\}$ for a point $\mathbf{p}'_k$ is defined as:

$$M_k = \mathbb{1}\left(z'_k = \min\{z'_m \mid \pi(\mathbf{p}'_m) = \pi(\mathbf{p}'_k)\}\right), \quad (3)$$

where the minimization is performed over all points $\mathbf{p}'_m$ that project to the same pixel coordinates as the target point $\mathbf{p}'_k$.

Consequently, the final correspondence map stores the coordinate mapping tuple $(\mathbf{x}_k, M_k)$ for every point. This mapping enables the precise back-projection of 2D anomaly scores to the 3D manifold during inference, effectively filtering out occluded noise.

### 3.3. Multi-View Attention Module

To prevent critical anomaly cues from being overshadowed by irrelevant background views, the Data-Driven Multi-View Attention (MVA) module adaptively aggregates 3D features by assigning learnable importance scores to each 2D view. By prioritizing semantic content over fixed weighting, MVA effectively suppresses occlusion noise and enhances the detection of localized anomalies, particularly in challenging zero-shot scenarios.

First, we employ a lightweight pixel encoder composed of a convolutional backbone and a projection head. Let $\mathcal{I} = \{\mathbf{I}_k\}_{k=1}^{N_v}$ denote the set of multi-view inputs consisting of $N_v$ rendered images. For the $k$-th view $\mathbf{I}_k$, the encoding process is formulated as a two-stage mapping. First, the convolutional backbone $\Phi_{\text{conv}}$ extracts spatial feature maps, which are then spatially aggregated via global average pooling $\mathcal{A}(\cdot)$. Subsequently, a linear projection head $\mathcal{W}$ maps the aggregated features to the final latent embedding $\mathbf{h}_k \in \mathbb{R}^d$:

$$\mathbf{h}_k = \mathcal{W}\left(\mathcal{A}\left(\Phi_{\text{conv}}(\mathbf{I}_k)\right)\right), \quad (4)$$

where $\Phi_{\text{conv}} : \mathbb{R}^{H \times W \times 3} \to \mathbb{R}^{H' \times W' \times C'}$ represents the stacked convolutional layers with $C'$ output channels, and $\mathcal{W} : \mathbb{R}^{C'} \to \mathbb{R}^d$ denotes the learnable linear transformation layer.

To adaptively weigh the contribution of each view, we employ a data-driven attention mechanism parameterized by a scoring network $\phi : \mathbb{R}^d \to \mathbb{R}$. This network projects the latent embedding $\mathbf{h}_k$ to a scalar unnormalized score $s_k = \phi(\mathbf{h}_k)$. We formulate the view importance as a normalized probability distribution $\boldsymbol{\alpha}$, such that $\alpha_k \geq 0$ and $\sum_{k=1}^{N_v} \alpha_k = 1$. To govern the sharpness of this distribution, we introduce a learnable temperature parameter $\tau$ and a scaling factor $\gamma$. The attention weight $\alpha_k$ is derived via the Boltzmann distribution:

$$\alpha_k = \frac{\exp(\gamma s_k / \tau)}{\sum_{j=1}^{N_v} \exp(\gamma s_j / \tau)}. \quad (5)$$

Consequently, the aggregated point cloud anomaly score $S_{point}$ is computed as the expectation of the view-specific predictions $S_{view}(\mathbf{I}_k)$ under the distribution $\boldsymbol{\alpha}$:

$$S_{point} = \mathbb{E}_{k \sim \boldsymbol{\alpha}}[S(\mathbf{I}_k)] = \sum_{k=1}^{N_v} \alpha_k S_{view}(\mathbf{I}_k). \quad (6)$$

Furthermore, to prevent the attention mechanism from collapsing into a trivial solution (e.g., always selecting a single view), we introduce an entropy regularization term $\mathcal{L}_{ent}$ during training to encourage distributional smoothness:

$$\mathcal{L}_{ent} = -\sum_{k=1}^{N_v} \alpha_k \log(\alpha_k + \epsilon), \quad (7)$$

where $\epsilon = 10^{-8}$ is a small constant for numerical stability. This loss term ensures that the distribution of weights remains informative while retaining the flexibility to focus on multiple relevant views when necessary.

### 3.4. Multi-Stage Color-Geometric Fusion Module

To effectively bridge the domain gap between 2D pre-trained features and 3D anomaly detection, we propose the Multi-Stage Color-Geometric Fusion (MS-CGF) module. This architecture systematically aggregates information across four dimensions: feature depth, semantic similarity, view, and modality.

**Feature Depth Aggregation.** Standard CLIP adaptation often relies solely on the final layer, overlooking fine-grained details critical for anomaly detection. To mitigate this, we partition the intermediate layers into $N_g$ groups. Let $\Omega_g = \{\mathbf{f}_1, \ldots, \mathbf{f}_L\}$ denote the set of feature maps in the $g$-th group, where $\mathbf{f}_L$ represents the deepest layer serving as the semantic reference. We employ an MLP-based Dynamic Feature Selector to compute the fused representation $\mathcal{F}_g$:

$$\mathcal{F}_g = \sum_{k=1}^{L} \boldsymbol{\beta}_k \cdot \mathbf{f}_k, \quad \text{with } \boldsymbol{\beta} = \sigma\left(\text{MLP}(\text{Avg}(\mathbf{f}_L))\right), \quad (8)$$

where $\sigma$ denotes the Softmax function, $\text{Avg}(\cdot)$ represents global average grouping, and $\boldsymbol{\beta} \in \mathbb{R}^L$ are the adaptive attention weights within the group.

**Semantic Similarity Computation.** Unlike previous methods that aggregate features before comparison, we perform anomaly detection at each semantic depth independently. For each feature group $g$, we compute the cosine similarity between the fused feature map $\mathcal{F}_g$ and the text embedding $\mathcal{T}$ derived from the prompt learner:

$$\mathcal{S}_g^{(i,j)} = \frac{\mathcal{F}_g^{(i,j)} \cdot \mathcal{T}^\top}{|\mathcal{F}_g^{(i,j)}||\mathcal{T}|}, \quad (9)$$

where $(i,j)$ denotes the pixel coordinate. Recognizing that different semantic levels contribute unequally, we introduce a learnable Weighted Sum mechanism to aggregate these maps from different groups. The unified view-wise anomaly map $\mathcal{S}_{view}$ is computed as:

$$\mathcal{S}_{view} = \sum_{g=1}^{N_g} \frac{\exp(w_g)}{\sum_{k=1}^{N_g} \exp(w_k)} \cdot \mathcal{S}_g, \quad (10)$$

where $w_g$ are the learnable parameters initialized to balance the contribution of shallow and deep features.

**View Aggregation.** Adaptive aggregation is then performed via a weighted summation of all $N_v$ views as Eq. 6, guided by MVA attention weights $\boldsymbol{\alpha}$. This mechanism prioritizes informative perspectives and suppresses occlusion noise, ensuring the final scores are robust to viewpoint variations.

**Modality Fusion.** Finally, we process the Color and Geometry streams in parallel. Let $S_{point}^{Co}$ and $S_{point}^{Geo}$ be the anomaly scores obtained for each modality; the final fused point cloud anomaly score $A_{point}$ is determined by an element-wise maximization strategy:

$$A_{point} = \max(S_{point}^{Co}, S_{point}^{Geo}). \quad (11)$$

This max-based fusion acts as a soft logical OR operation allowing prominent anomaly evidence from either domain to dominate. By avoiding the signal dilution inherent in averaging, this approach maintains superior sensitivity to both textural and structural defects.

### 3.5. Loss Design

To enable accurate localization and robust recognition in both 2D and 3D spaces, we adopt a unified multi-task learning objective. The overall loss $\mathcal{L}_{total}$ jointly incorporates segmentation supervision, classification supervision, and attention regularization. For fine-grained localization, we impose constraints on both the rendered view scores and the back-projected 3D manifold. For the views, we address the inherent class imbalance between sparse anomalous pixels and dominant normal pixels by combining Focal Loss and Dice Loss:

$$\mathcal{L}_{seg}^{view} = \mathcal{L}_{Focal}(S_{view}, M_{gt}) + \mathcal{L}_{Dice}(S_{view}, M_{gt}), \quad (12)$$

where $S_{view}$ is the aggregated similarity map and $M_{gt}$ is the corresponding ground-truth mask. The Focal Loss term focuses on hard-to-classify boundary pixels, while the Dice Loss ensures structural overlap between the predicted and actual anomaly regions.

After back-projecting the view scores, we apply a Binary Dice Loss on the point cloud to ensure structural alignment with the 3D ground truth:

$$\mathcal{L}_{seg}^{point} = \mathcal{L}_{Dice}(A_{point}, P_{gt}), \quad (13)$$

where $A_{point} \in \mathbb{R}^N$ represents the final point-wise anomaly scores and $P_{gt}$ denotes the 3D ground-truth labels. This point-specific constraint filters out projection noise and ensures that detected anomalies conform to the physical object surface.

To enhance the discriminative power, we implement a hierarchical supervision strategy using Binary Cross-Entropy

(BCE) loss. Let $\ell_{bce}(p, y)$ denote the standard BCE objective, where $y \in \{0, 1\}$ is the ground truth label. First, we introduce a view-wise classification loss $\mathcal{L}_{cls}^{view}$ that independently supervises each view $k$. Let $p_k$ denote the predicted probability for view $k$; the loss is averaged over all views:

$$\mathcal{L}_{cls}^{view} = \frac{1}{N_v} \sum_{k=1}^{N_v} \ell_{bce}(p_k, y). \qquad (14)$$

Similarly, we apply the same objective to the aggregated point-level prediction $p_{agg}$ to ensure global consistency:

$$\mathcal{L}_{cls}^{point} = \ell_{bce}(p_{agg}, y). \qquad (15)$$

Finally, we incorporate an entropy loss $\mathcal{L}_{\text{ent}}$ on the MVA weights to remain informative in the rendered views. The total loss $\mathcal{L}_{total}$ is defined as:

$$\mathcal{L}_{total} = \mathcal{L}_{seg}^{view} + \mathcal{L}_{seg}^{point} + \mathcal{L}_{cls}^{view} + \mathcal{L}_{cls}^{point} + \mathcal{L}_{\text{ent}}. \quad (16)$$

## 4. Experiment

### 4.1. Dataset

We evaluate the CoGeoAD model using the MVTec 3D-AD (Bergmann et al., 2021) and Eyecandies (Bonfiglioli et al., 2022) datasets, each comprising ten distinct categories with geometric point clouds (depth maps) and RGB images. We primarily assess performance through cross-dataset zero-shot generalization, where the model is trained on one dataset and tested on the other. To comprehensively evaluate CoGeoAD, we utilize four metrics: the Area Under the Receiver Operating Characteristic curve (AUROC) and Average Precision (AP) for image-level (I-AUROC, AP) and pixel-level (P-AUROC) tasks. Additionally, we compute Area Under the Per-Region Overlap (AUPRO) to assess the spatial overlap between predicted anomaly maps and ground truth masks, ensuring robustness to variations in anomaly size. We further report results on the Real3D-AD (Liu et al., 2023) dataset in Appendix C.

### 4.2. Implementation Details

The image encoder and text encoder are adopted from the pre-trained CLIP model (ViT-L/14@336px). Consistent with the visual backbone, the network takes $336 \times 336$ three-channel images as input, producing 768-dimensional embedding features. We render the point cloud into multi-view 2D images by sequentially rotating the object around the X and Y axes. Specifically, we define a set of viewing angles with X-axis rotations $\theta_x \in \{-\frac{\pi}{4}, -\frac{\pi}{12}, 0, \frac{\pi}{12}, \frac{\pi}{4}\}$ and Y-axis rotations $\theta_y \in \{-\frac{\pi}{6}, -\frac{\pi}{12}, \frac{\pi}{12}, \frac{\pi}{6}\}$. This combination generates a total pool of 20 candidate views. To balance computational efficiency with geometric coverage, we select $N_v = 9$ views from this generated sequence for model training.

All experiments are conducted on a single NVIDIA RTX 3090 GPU (24GB) utilizing PyTorch-2.0.0. To preserve pre-trained knowledge, the CLIP backbone parameters remain frozen throughout the process. Optimization is focused exclusively on the newly introduced modules: the Prompt Learner, the View Attention module, and the Multi-Stage Color-Geometric Fusion Module. Training is performed using the Adam optimizer with a learning rate of $1 \times 10^{-3}$, betas of $(0.5, 0.999)$, and a batch size of 4 over 20 epochs.

### 4.3. Main Results

In this section, we evaluate CoGeoAD on the MVTec3D-AD (Bergmann et al., 2021) and Eyecandies (Bonfiglioli et al., 2022) datasets, comparing it against state-of-the-art (SOTA) zero-shot methods, including PointCLIP V2 (Zhu et al., 2023), MVP-PCLIP (Cheng et al., 2025b), AnomalyCLIP (Zhou et al., 2023), PointAD (Zhou et al., 2024a), PointAD+ (Zhou et al., 2025), and the latest CLIP-based GS-CLIP (Deng et al., 2026). We also include strong unsupervised multimodal methods, including CPMF (Cao et al., 2024a) and BridgeNet (Xiang et al., 2025), as reference upper bounds. Recent 2025 methods such as PASDF (Zheng et al., 2025), DUS-Net (Liang et al., 2025b), MC3D-AD (Cheng et al., 2025a), Simple3D/MiniShift (Cheng et al., 2026), and MuSc-V2 (Li et al., 2025) are discussed in Related Work; we do not mix their results into Table 1 when they are reported on different datasets or under different metric protocols. Detailed per-class performance comparisons on both MVTec3D-AD and Eyecandies datasets are provided in Appendix B.

As summarized in Table 1, CoGeoAD establishes a new SOTA among directly comparable zero-shot 3D anomaly detection methods. On MVTec3D-AD, our method achieves the best zero-shot I-AUROC (87.4%), P-AUROC (97.5%), and AUPRO (91.9%), while matching the strongest AP score (96.5%). Compared with the latest GS-CLIP baseline, CoGeoAD improves I-AUROC by 3.8%, P-AUROC by 1.2%, and AUPRO by 5.5%, showing the benefit of hierarchical color-geometric fusion over geometry-aware prompting alone. Although unsupervised methods such as CPMF and BridgeNet can obtain strong results by relying on target-domain normal training data or memory-based modeling, CoGeoAD operates in a zero-shot setting without target dataset adaptation. On the Eyecandies dataset, CoGeoAD consistently outperforms all zero-shot competitors across all metrics, surpassing PointAD+ by 2.3% in I-AUROC, 1.4% in AP, 2.4% in P-AUROC, and 2.0% in AUPRO, and exceeding GS-CLIP by 12.3% I-AUROC and 13.8% AUPRO. These gains demonstrate the robustness of CoGeoAD in handling complex object geometries and textures.

Beyond detection accuracy, we benchmark the computational efficiency of CoGeoAD in Table 2. Despite the intro-

*Table 1.* Quantitative comparison with state-of-the-art methods on MVTec3D-AD and Eyecandies datasets. We include recent unsupervised multimodal methods such as BridgeNet as reference upper bounds, and recent zero-shot CLIP-based methods such as GS-CLIP for direct comparison. Within the zero-shot category, the best results are highlighted in **bold**, and the second-best results are underlined. "-" indicates the results are not reported under the same benchmark or metric protocol.

| Method | Venue | MVTec3D-AD | | | | Eyecandies | | | |
|---|---|---|---|---|---|---|---|---|---|
| | | I-AUROC | AP | P-AUROC | AUPRO | I-AUROC | AP | P-AUROC | AUPRO |
| BTF(Unsupervised) | CVPR'23 | 86.5 | - | 99.2 | 95.9 | 82.1 | - | - | 84.6 |
| M3DM(Unsupervised) | CVPR'23 | 94.5 | - | 99.2 | 96.4 | 89.7 | - | - | 88.2 |
| CPMF(Unsupervised) | CVPR'23 | 95.2 | 98.2 | 93.5 | 92.9 | - | - | - | - |
| BridgeNet(Unsupervised) | MM'25 | 99.3 | - | 99.6 | 97.7 | 95.8 | - | - | 92.9 |
| CLIP+Render | - | 60.4 | 86.4 | - | 56.0 | 73.0 | 73.9 | 78.0 | 31.8 |
| PointCLIP V2 | ICCV'23 | 78.3 | 49.4 | 87.4 | - | 48.5 | 50.9 | 46.3 | - |
| PointCLIP V2a | ICCV'23 | 49.4 | 79.8 | 48.5 | 50.5 | 48.5 | 50.5 | 46.2 | - |
| MVP-PCLIP | T-SMC'25 | 71.9 | 89.6 | 87.5 | - | 54.8 | 56.7 | 72.2 | - |
| AnomalyCLIP | ICLR'24 | 65.3 | 87.6 | 92.5 | 74.7 | 62.4 | 62.4 | 89.8 | 69.7 |
| PointAD | NeurIPS'24 | 78.9 | 93.2 | 94.3 | 80.6 | 77.2 | 79.9 | 94.5 | 83.6 |
| PointAD+ | arXiv'25 | 82.5 | 94.8 | 96.9 | 88.7 | 80.3 | 83.1 | 94.8 | 85.2 |
| GS-CLIP | CVPR'26 | 83.6 | **96.5** | 96.3 | 86.4 | 70.3 | 75.3 | 92.9 | 73.4 |
| **CoGeoAD (Ours)** | - | **87.4** | **96.5** | **97.5** | **91.9** | **82.6** | **84.5** | **97.2** | **87.2** |

*Table 2.* Comparison of computational efficiency and performance on MVTec3D-AD.

| Method | GPU memory usage (Peak) | GFLOPs | FPS | I-AUROC | P-AUROC |
|---|---|---|---|---|---|
| CLIP+Render | 3685MB | 1057.36 | 3.61 | 60.4 | - |
| PointCLIP V2 | 9747MB | 4660.49 | 0.66 | 78.3 | 87.4 |
| MVP-PCLIP | 5694MB | **467.35** | **8.03** | 71.9 | 87.5 |
| AnomalyCLIP | **3348MB** | 1656.34 | 5.26 | 65.3 | 92.5 |
| PointAD | 4275MB | 1656.34 | 2.52 | 78.9 | 94.3 |
| PointAD+ | 5811MB | 1658.34 | 2.18 | 82.5 | 96.9 |
| CoGeoAD (Ours) | 4445MB | 3305.65 | 1.28 | **87.4** | **97.5** |

duction of the Multi-View Attention (MVA) mechanism, Co-GeoAD achieves a superior trade-off between performance and cost. Compared to the heavy-weight PointCLIP V2, our method reduces GFLOPs by approximately 29.1% (3305.65 vs. 4660.49) and nearly doubles the inference throughput (1.28 FPS vs. 0.66 FPS). Furthermore, CoGeoAD's peak memory footprint of 4445MB is significantly lower than that of PointCLIP V2 (9747MB), MVP-PCLIP (5694MB), and PointAD+ (5811MB), making it highly suitable for deployment on consumer-grade GPUs. While lightweight baselines like AnomalyCLIP or PointAD offer higher FPS, the moderate computational overhead of CoGeoAD is justified by its substantial gains in detection precision.

To intuitively demonstrate the effectiveness of our dual-branch design, Fig. 3 illustrates the complementarity of our dual-branch design. The Color Branch accurately identifies textural stains (e.g., Foam), while the Geometric Branch captures structural cracks (e.g., Bagel). The final fusion effectively integrates these cues, ensuring robust detection across diverse anomaly types.

### 4.4. Ablation Study

We analyze the impact of each module and various hyperparameters through a series of ablations.

**Effectiveness of Key Components.** Table 3 demonstrates the necessity of our proposed modules. The exclusion of the **Color** or **Geometric** branch leads to a noticeable performance decline, confirming that 3D anomaly detection benefits from multi-modal synergy. Furthermore, removing the Multi-Stage Color-Geometric Fusion (MS-CGF) drops the AUPRO from 91.9% to 87.9% on MVTec3D-AD, proving its effectiveness in capturing multi-level anomalies.

**Impact of Multi-View Attention (MVA).** To isolate the contribution of MVA, we provide quantitative and qualitative ablations in Table 4 and Fig. 4. Replacing MVA with standard Mean Pooling results in a 3.1% drop in I-AUROC. As visualized in Fig. 4, MVA intelligently assigns higher weights to "informative" views (e.g., View 8, where the defect is prominent) while suppressing noisy or occluded perspectives (e.g., View 1). This dynamic prioritization yields cleaner 3D anomaly maps and more precise localization compared to vanilla Average Fusion.

**Impact of View Number and Resolution.** Table 5 shows that performance peaks at $N_v = 9$ views. Increasing views further to 12 introduces redundancy and increases latency significantly. Additionally, high resolution is vital; decreasing from 336 to 224 causes a sharp drop in I-AUROC (from 87.4% to 75.1%), as low-resolution images struggle to resolve subtle geometric anomalies.

**Layer Fusion Strategy.** We investigate the impact of different layer combinations from the CLIP ViT-L/14 backbone

*Table 3.* Ablation study on key components of CoGeoAD on MVTec3D-AD and Eyecandies datasets. Within the zero-shot category, the best results are highlighted in **bold**, and the second-best results are underlined. "-" indicates the results are not reported.

| Module | | | | MVTec3D-AD | | | | Eyecandies | | | |
|---|---|---|---|---|---|---|---|---|---|---|---|
| Color | Geo | MVA | MS-CGF | I-AUROC | AP | P-AUROC | AUPRO | I-AUROC | AP | P-AUROC | AUPRO |
| × | ✓ | ✓ | - | 84.3 | 95.0 | 97.1 | 89.6 | 81.0 | 83.4 | 95.9 | 83.7 |
| ✓ | × | ✓ | - | 83.6 | 95.2 | 97.3 | 91.3 | 79.7 | 82.5 | 96.2 | 81.8 |
| ✓ | ✓ | × | ✓ | 86.2 | 96.0 | 97.4 | 90.8 | 78.9 | 81.3 | 95.9 | 83.9 |
| ✓ | ✓ | ✓ | × | 83.5 | 95.2 | 96.4 | 87.9 | 82.3 | 84.4 | 97.0 | 85.9 |
| ✓ | ✓ | ✓ | ✓ | **87.4** | **96.5** | **97.5** | **91.9** | **82.6** | **84.5** | **97.2** | **87.2** |

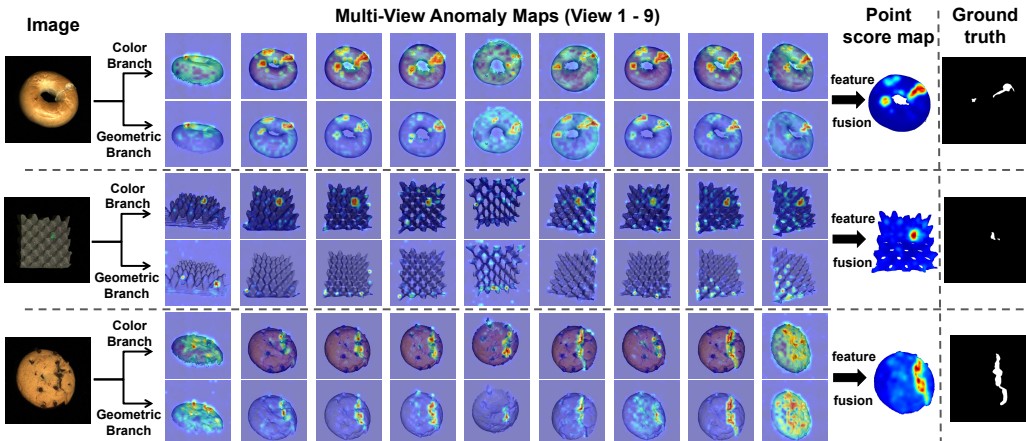

*Figure 3.* Visualization of complementary feature learning. The Geometric Branch captures structural cracks in the top row, while the Color Branch identifies textural stains in the middle row. The final fused map integrates these complementary cues via our MS-CGF module for accurate 3D localization.

*Table 4.* Ablation on Layer Fusion and View Fusion Methods on the MVTec3D-AD dataset.

| Setting | I-AUROC | AP | P-AUROC | AUPRO |
|---|---|---|---|---|
| *Layer Fusion Strategy* | | | | |
| Shallow (Layers 1–12) | 50.9 | 80.2 | 93.7 | 79.4 |
| All (Layers 1–24) | 80.6 | 94.1 | 96.4 | 87.6 |
| Deep (Layers 13–24) | 84.8 | 95.6 | 97.2 | 90.3 |
| Strided (Layers 6, 12, 18, 24) | 86.8 | 96.2 | 97.4 | 91.3 |
| **Last (Layers 21–24)** | **87.4** | **96.5** | **97.5** | **91.9** |
| *View Fusion Method based on Last Layers 21–24* | | | | |
| Mean Pooling | 83.8 | 95.3 | 96.2 | 88.2 |
| Static Scalars | 82.3 | 94.6 | 95.4 | 85.1 |
| MLP Weighting | 85.0 | 95.7 | 97.2 | 90.8 |
| **MVA (Ours)** | **87.4** | **96.5** | **97.5** | **91.9** |

*Table 5.* Impact of image resolution and view numbers on the MVTec3D-AD dataset.

| Resolution | Views | I-AUROC | AP | P-AUROC | AUPRO | FPS |
|---|---|---|---|---|---|---|
| 336 × 336 | 2 | 81.7 | 93.9 | 96.2 | 87.2 | 2.70 |
| | 4 | 85.1 | 95.5 | 97.4 | 90.9 | 2.49 |
| | 6 | 84.8 | 95.4 | 97.3 | 90.3 | 1.78 |
| | 8 | 86.2 | 96.1 | **97.5** | 91.0 | 1.38 |
| | **9** | **87.4** | **96.5** | **97.5** | **91.9** | 1.28 |
| | 10 | 85.0 | 95.6 | 97.2 | 90.4 | 1.17 |
| | 12 | 82.1 | 94.5 | 96.4 | 88.6 | 0.98 |
| 224 × 224 | 6 | 72.9 | 91.1 | 93.8 | 80.8 | 3.77 |
| | 9 | 75.1 | 91.8 | 94.6 | 83.3 | 2.62 |
| | 12 | 80.4 | 93.8 | 95.3 | 85.5 | 2.15 |

in Table 4. Our results reveal a clear performance trend: deeper layers are significantly more effective for zero-shot anomaly detection. Specifically, using only Shallow layers 1-12 yields a near-random I-AUROC (50.9%), whereas the Last four layers 21-24 achieve the peak performance of 87.4%. This suggests that while shallow layers capture low-level textures, the high-level semantic abstractions in the final layers are crucial for defining "normality" and identifying deviations in a zero-shot context. Notably, our Last strategy outperforms the Strided approach, indicating that dense information from the tail of the network is most

representative for this task.

**View Fusion Method.** We compare our proposed Multi-View Attention (MVA) against several baseline fusion strategies. As shown in the lower half of Table 4, MVA consistently outperforms all alternatives. Compared to standard Mean Pooling, MVA improves I-AUROC by 3.1% and AUPRO by 3.5%, demonstrating that treating all views equally introduces noise from uninformative or occluded perspectives. While MLP Weighting and Static Scalars offer marginal improvements over mean pooling, they lack

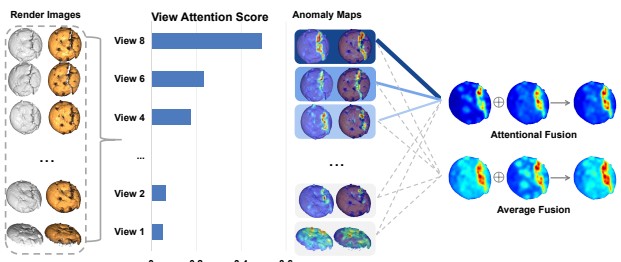

*Figure 4.* Visualization of the Data-Driven Multi-View Attention (MVA) mechanism.

*Table 6.* Detailed quantitative evaluation across different anomaly size intervals. The metrics confirm that our CoGeoAD framework reliably detects micro-scale anomalies.

| Interval (pts) | P-AUROC | AUPRO | I-AUROC | AP |
|---|---|---|---|---|
| 1–100 | 97.8 | 91.7 | 79.7 | 69.5 |
| 101–500 | 98.2 | 92.9 | 87.5 | 92.9 |
| 501–1000 | 98.6 | 94.8 | 90.5 | 91.6 |
| 1001–2000 | 99.0 | 96.8 | 92.9 | 78.5 |
| 2001–5000 | 95.2 | 88.6 | 95.9 | 95.8 |
| 5000+ | 93.0 | 84.5 | 100.0 | 100.0 |

the dynamic adaptability of MVA. MVA's ability to assign instance-specific weights based on visual content ensures that the most prominent defect features are prioritized, leading to the highest P-AUROC (97.5%) and AUPRO (91.9%).

**Impact of Anomaly Size.** We analyze the detection and localization performance of CoGeoAD across varying ground-truth anomaly sizes, as detailed in Table 6. Predictably, global image-level metrics decline for micro-scale anomalies due to the weak global signal they exhibit, with I-AUROC dropping to 79.7% in the smallest interval (1–100 points) before perfectly scaling to 100.0% for anomalies larger than 5000 points. However, our method demonstrates exceptional robustness in precise defect localization regardless of scale. Notably, local metrics remain incredibly stable, achieving a highly competitive 97.8% P-AUROC and 91.7% AUPRO even for the extremely challenging 1–100 point interval. This resilience highlights the effectiveness of our 3D-to-2D point-to-pixel alignment strategy, wherein high-density 3D point cloud sampling successfully compensates for 2D resolution limits, enabling accurate localization of sub-millimeter defects.

**Robustness to Noise and Calibration Shifts.** Table 7 evaluates the resilience of CoGeoAD against simulated "in-the-wild" perturbations, specifically point cloud Gaussian noise(GN) and camera calibration shifts (pixel misalignments). The model exhibits remarkable stability when subjected to isolated perturbations; for instance, introducing significant calibration shifts of up to $\pm 10$px results in a negligible drop in I-AUROC (from 87.4% to 87.2%), and Gaussian noise up to std=0.1 similarly causes minimal degra-

*Table 7.* Robustness analysis against point cloud noise and camera calibration shifts (pixel misalignments). Our CoGeoAD demonstrates strong resilience to both independent and mixed perturbations, maintaining high performance even under severe "in-the-wild" noise simulations.

| Perturbation Type | GN | Calib-Shift | I-AUROC | AP | P-AUROC | AUPRO |
|---|---|---|---|---|---|---|
| Ideal Condition | std=0 | $\pm$0px | **87.4** | **96.5** | **97.5** | **91.9** |
| Gaussian Noise | std=0.05 | $\pm$0px | 87.0 | 96.3 | 97.7 | 91.9 |
| | std=0.1 | $\pm$0px | 86.8 | 96.2 | 97.6 | 91.9 |
| | std=0.2 | $\pm$0px | 84.5 | 95.4 | 97.4 | 91.5 |
| Calibration Shift | std=0 | $\pm$5px | 87.2 | 96.4 | 97.5 | 91.3 |
| | std=0 | $\pm$10px | 87.2 | 96.4 | 97.4 | 91.1 |
| Mix | std=0.05 | $\pm$5px | 86.9 | 96.3 | 97.6 | 91.7 |
| | std=0.1 | $\pm$10px | 87.0 | 96.3 | 97.5 | 91.5 |
| | std=0.2 | $\pm$10px | 84.3 | 95.3 | 97.3 | 90.7 |
| | std=0.5 | $\pm$20px | 78.9 | 93.4 | 96.7 | 88.7 |

dation. More importantly, CoGeoAD demonstrates strong robustness when confronting mixed perturbations. Even under extreme conditions (std=0.5 noise combined with a $\pm 20$px shift), the model degrades gracefully, retaining a competitive P-AUROC of 96.7% and an AUPRO of 88.7%. These results confirm that our approach does not strictly rely on pristine point clouds or perfect sensor calibration, making it highly practical and reliable for complex, real-world industrial deployments.

## 5. Conclusion

In this work, we introduced CoGeoAD, a zero-shot 3D industrial anomaly detection framework that integrates 2D semantic priors with 3D geometric representations. By employing the Data-Driven Multi-View Attention (MVA) module, CoGeoAD adaptively prioritizes salient viewpoints, effectively mitigating noise inherent in conventional pooling-based aggregation. Specifically, we implement a dual-stream rendering strategy that explicitly decouples geometric structures from surface textures by generating complementary chromatic and geometric representations. Furthermore, the Multi-Stage Color-Geometric Fusion (MS-CGF) module enables hierarchical feature alignment, ensuring precise localization of both textural and structural anomalies. While CoGeoAD achieves state-of-the-art performance on the MVTec 3D-AD and Eyecandies benchmarks, it reveals a critical trade-off between geometric coverage and computational efficiency. Future work will address the observed redundancy in high-density view sampling and further refine the semantic alignment between specialized industrial defects and general-purpose vision-language models.

## Acknowledgments

This work was supported in part by the National Natural Science Foundation of China (No. 62576001, No. 62206003), the Natural Science Foundation of Anhui Province (2308085MF201), the Key Program of Natural

Science Project of the Educational Commission of Anhui Province (KJ2021A0048), and the Major Project of Natural Science Research, Anhui Provincial Department of Education (2025AHGXZK20062).

## Impact Statement

This work advances zero-shot 3D anomaly detection for industrial inspection, which may reduce annotation costs and assist quality-control workflows. In practical deployment, false negatives may allow defective products to escape inspection, while false positives may increase manual review costs. Therefore, CoGeoAD is intended as a decision-support or pre-screening tool rather than a replacement for human inspectors in safety-critical manufacturing scenarios. We do not identify additional societal impacts beyond these deployment considerations.

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

# A. Detailed Qualitative Visualizations

In this appendix, we provide a comprehensive visualization of the CoGeoAD inference process on the MVTec3D-AD and Eyecandies datasets, demonstrating how our Data-Driven Multi-View Attention mechanism adaptively aggregates 3D features by displaying the original RGB image, the anomaly score maps for all 9 rendered views, the final fused 3D point-wise anomaly score map, and the ground truth annotations for each sample. Figure 5 and Figure 6 illustrate the complete breakdown of this process, further validating the effectiveness of the Multi-Stage Color-Geometric Fusion module in bridging the 2D-3D domain gap and localizing defects that may be occluded in specific single views.

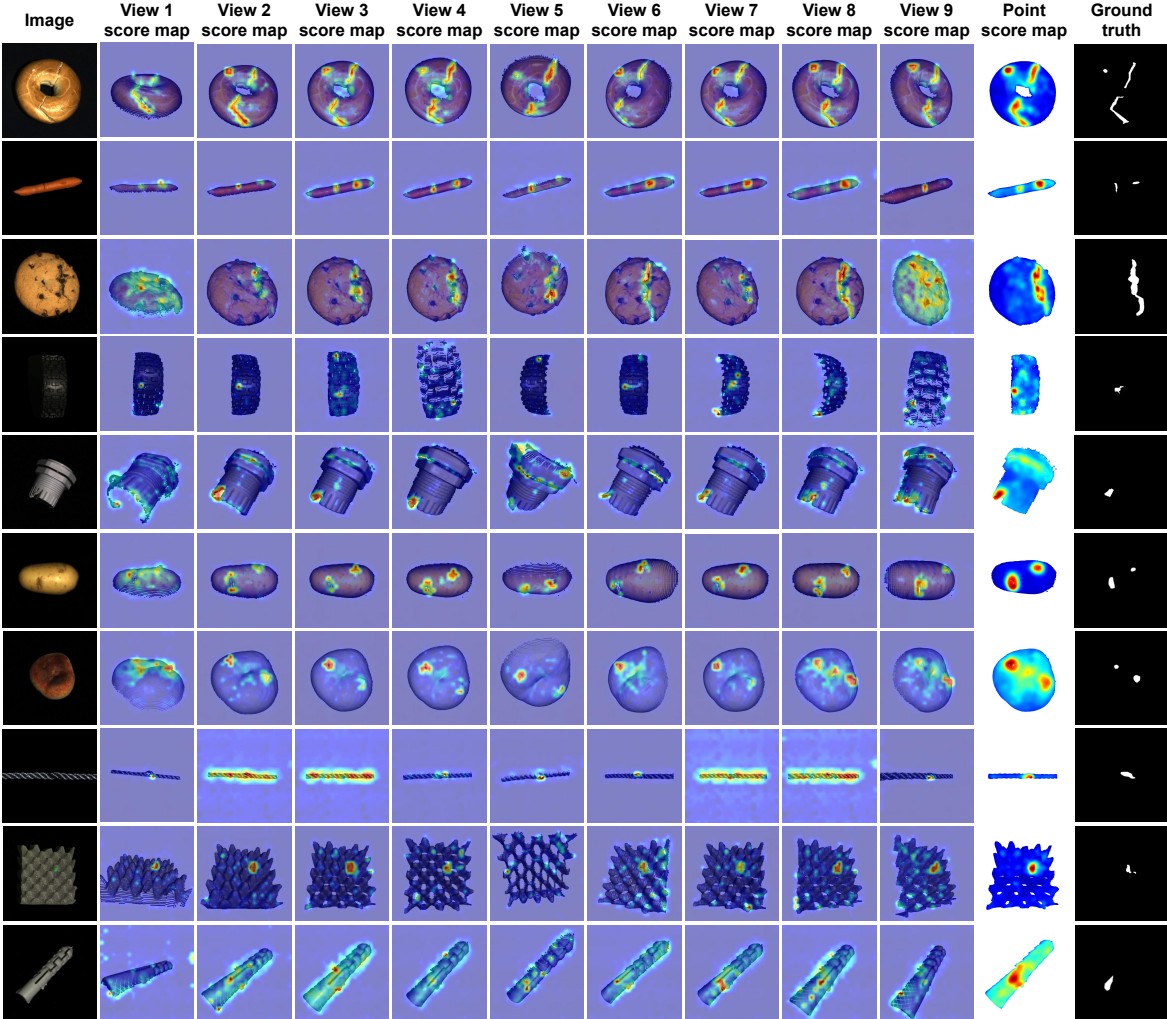

*Figure 5.* Detailed qualitative results of CoGeoAD on the MVTec3D-AD dataset, showing the view-wise decomposition of anomaly detection from the input RGB image through individual rendered views to the final fused 3D score map and ground truth.

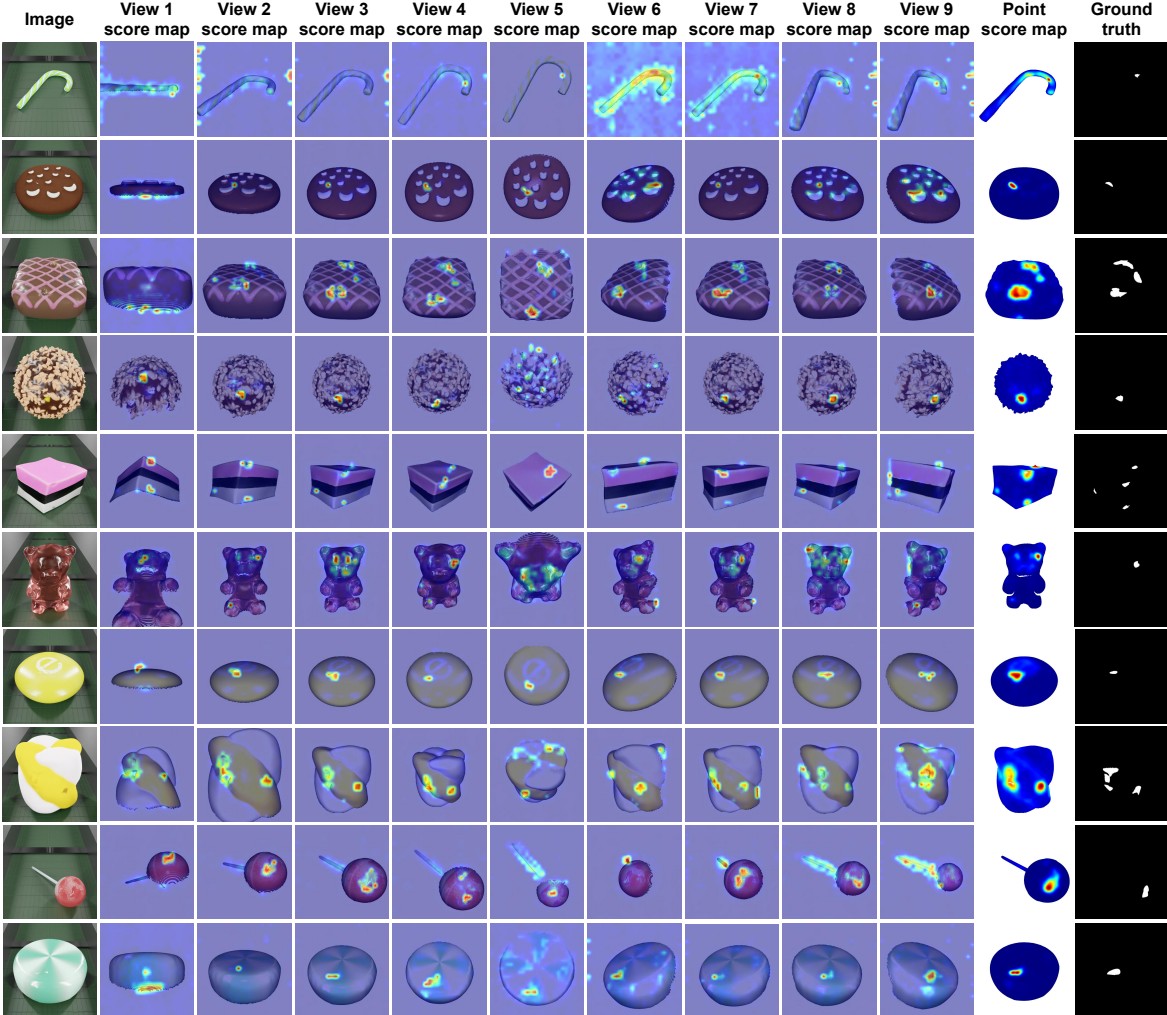

*Figure 6.* Detailed qualitative results of CoGeoAD on the Eyecandies dataset, illustrating the robust performance of the multi-view aggregation and fusion modules across various complex object geometries and textures.

## B. Detailed Quantitative Results

In this section, we present the fine-grained quantitative results for each category on the MVTec3D-AD and Eyecandies datasets. For image-level anomaly detection, we report the Area Under the Receiver Operating Characteristic curve (AUC) and Average Precision (AP), while for pixel-level anomaly localization, we report the pixel-wise AUC and the Area Under the Per-Region Overlap (AUPRO). Table 8 summarizes the results on the MVTec3D-AD dataset for image-level and pixel-level tasks, respectively, whereas Table 9 presents the corresponding results on the Eyecandies dataset. In all tables, the results are presented in the format of *(Metric 1, Metric 2)*, corresponding to *((I-AUROC, AP)* for image-level and *((P-AUROC, AUPRO)* for pixel-level results, where the best performance in each category is highlighted in **bold** and the second-best is indicated by an underline.

## C. Additional Results on Real3D-AD

We additionally evaluate CoGeoAD on the Real3D-AD dataset to assess its generalization beyond MVTec3D-AD and Eyecandies. As shown in Table 10, CoGeoAD achieves the best mean performance across all three metrics, reaching 90.6% P-AUROC, 79.9% I-AUROC, and 81.7% I-AP.

*Table 8.* Quantitative results on the MVTec3D-AD dataset (%).

| | Method | Bagel | Cable_gland | Carrot | Cookie | Dowel | Foam | Peach | Potato | Rope | Tire | Mean |
|---|---|---|---|---|---|---|---|---|---|---|---|---|
| | PointCLIP V2 | (51.6, 83.7) | (63.8, 87.6) | (47.7, 83.5) | (47.8, 78.0) | (51.8, 80.5) | (45.2, 78.5) | (49.2, 78.7) | (55.4, 82.9) | (39.1, 62.4) | (46.0, 76.9) | (49.8, 79.3) |
| | PointCLIP V2a | (53.4, 84.4) | (64.7, 89.1) | (48.0, 83.4) | (48.4, 78.4) | (47.1, 81.1) | (45.9, 79.0) | (49.6, 79.2) | (55.5, 85.9) | (34.9, 60.5) | (46.1, 76.9) | (49.4, 79.8) |
| | MVP-PCLIP | (43.6, 80.7) | (48.9, 80.1) | (58.6, 88.5) | (64.6, 87.7) | (40.8, 76.7) | (31.9, 72.0) | (53.2, 83.6) | (65.6, 89.0) | (32.4, 63.0) | (47.4, 78.5) | (48.7, 80.0) |
| Image-level | AnomalyCLIP | (81.2, 94.7) | (65.7, 89.8) | (70.5, 92.2) | (68.5, 90.2) | (58.7, 86.2) | (67.8, 91.2) | (76.2, 92.9) | (66.7, 90.7) | (38.8, 64.7) | (59.2, 82.8) | (65.3, 87.6) |
| (AUROC, AP) | PointAD | (90.1, 97.4) | (64.9, 89.2) | (94.7, 98.9) | (67.4, 89.8) | (73.0, 93.0) | (70.2, 90.3) | (91.0, 97.5) | (90.4, 96.3) | (86.8, 94.6) | (60.0, 85.5) | (78.9, 93.2) |
| | CoGeoAD | (97.6, 99.4) | (69.2, 91.7) | (97.7, 99.4) | (89.1, 96.9) | (75.6, 93.4) | (84.1, 96.0) | (94.2, 98.3) | (97.6, 99.4) | (96.7, 98.7) | (66.9, 89.9) | (87.4, 96.5) |
| | PointCLIP V2 | (40.6, 78.0) | (56.1, 84.4) | (53.8, 84.2) | (52.7, 81.1) | (50.7, 80.4) | (40.8, 78.1) | (54.9, 82.8) | (48.9, 77.9) | (54.3, 72.5) | (59.3, 81.9) | (78.3, 49.4) |
| | PointCLIP V2a | (75.9, 40.8) | (76.2, 47.4) | (92.5, 79.9) | (71.7, 30.7) | (72.8, 44.9) | (62.3, 21.9) | (77.1, 46.4) | (87.4, 63.7) | (87.9, 69.9) | (90.8, 70.8) | (79.5, 51.6) |
| Pixel-level | MVP-PCLIP | (95.3, -) | (77.8, -) | (94.8, -) | (87.7, -) | (81.7, -) | (67.7, -) | (97.4, -) | (97.7, -) | (96.8, -) | (84.0, -) | (88.1, -) |
| (AUROC, PRO) | AnomalyCLIP | (89.1, 64.2) | (94.6, 78.7) | (98.0, 92.9) | (82.5, 48.6) | (93.6, 78.0) | (89.6, 64.1) | (92.5, 72.9) | (97.8, 91.0) | (94.1, 80.7) | (93.3, 76.0) | (92.5, 74.7) |
| | PointAD | (95.0, 82.0) | (95.3, 83.0) | (99.1, 97.2) | (87.6, 64.1) | (94.3, 76.7) | (86.1, 51.8) | (96.4, 88.6) | (99.2, 95.9) | (97.1, 88.9) | (93.1, 77.7) | (94.3, 80.6) |
| | CoGeoAD | (99.6, 98.6) | (96.4, 86.0) | (99.7, 98.9) | (93.2, 87.5) | (96.2, 85.1) | (93.7, 78.9) | (99.4, 98.3) | (99.9, 99.2) | (99.3, 95.4) | (97.7, 89.0) | (97.5, 91.9) |

*Table 9.* Quantitative results on the Eyecandies dataset (%).

| | Method | Candy Cane | Chocolate Cookie | Chocolate Praline | Confetto | Gummy Bear | Hazelnut Truffle | Licorice Sandwich | Lollipop | Marshmallow | Peppermint Candy | Mean |
|---|---|---|---|---|---|---|---|---|---|---|---|---|
| | PointCLIP V2 | (43.0, 48.3) | (48.0, 55.0) | (46.4, 51.6) | (49.3, 48.4) | (44.7, 49.1) | (48.3, 55.4) | (61.8, 70.0) | (42.1, 30.4) | (54.1, 51.5) | (31.2, 39.6) | (46.9, 49.9) |
| | PointCLIP V2a | (44.1, 51.2) | (44.5, 52.4) | (48.6, 52.3) | (56.7, 54.8) | (42.8, 44.6) | (55.7, 60.3) | (63.5, 68.7) | (43.8, 29.3) | (54.0, 52.1) | (31.3, 39.6) | (48.5, 50.5) |
| | MVP-PCLIP | (51.4, 55.1) | (49.1, 50.2) | (69.0, 74.0) | (75.5, 80.4) | (59.2, 59.1) | (63.2, 71.8) | (73.1, 79.4) | (52.2, 42.2) | (49.4, 55.8) | (42.2, 54.9) | (58.4, 62.3) |
| Image-level | AnomalyCLIP | (43.4, 50.9) | (66.4, 64.5) | (73.6, 71.5) | (68.3, 71.1) | (51.5, 51.6) | (58.2, 57.2) | (60.5, 67.1) | (57.4, 34.6) | (82.9, 89.3) | (61.4, 65.9) | (62.4, 62.4) |
| (AUROC, AP) | PointAD | (47.7, 53.5) | (85.4, 89.3) | (82.9, 88.4) | (95.0, 96.7) | (72.0, 73.6) | (65.8, 68.0) | (88.8, 90.7) | (72.8, 68.6) | (76.6, 80.8) | (85.0, 89.1) | (77.2, 79.9) |
| | CoGeoAD | (49.3, 49.2) | (85.6, 90.4) | (94.1, 96.4) | (90.9, 92.9) | (84.0, 87.6) | (77.4, 81.9) | (90.1, 92.7) | (73.9, 66.7) | (93.1, 95.7) | (87.4, 91.4) | (82.6, 84.5) |
| | PointCLIP V2 | (44.8, -) | (44.8, -) | (48.0, -) | (59.6, -) | (48.6, -) | (53.9, -) | (42.2, -) | (33.7, -) | (43.3, -) | (41.4, -) | (46.0, -) |
| | PointCLIP V2a | (44.8, -) | (44.7, -) | (49.0, -) | (59.3, -) | (48.2, -) | (54.2, -) | (42.2, -) | (33.6, -) | (45.0,-) | (41.5, -) | (46.2, -) |
| Pixel-level | MVP-PCLIP | (83.8, -) | (79.1, -) | (81.5, -) | (91.2, -) | (71.4, -) | (77.0, -) | (79.7, -) | (81.7, -) | (77.1, -) | (86.2, -) | (80.9, -) |
| (AUROC, PRO) | AnomalyCLIP | (95.6, 84.7) | (89.1, 66.2) | (93.8, 81.1) | (95.6, 84.0) | (86.7, 58.9) | (81.6, 48.5) | (84.3, 61.8) | (93.3, 78.4) | (85.9, 55.3) | (92.2, 78.6) | (89.8, 69.7) |
| | PointAD | (97.2, 91.0) | (97.4, 89.6) | (94.3, 84.0) | (97.7, 93.9) | (90.0, 69.8) | (82.9, 59.4) | (97.6, 91.5) | (97.8, 89.5) | (93.9, 81.5) | (96.2, 85.6) | (94.5, 83.6) |
| | CoGeoAD | (94.3, 81.0) | (97.9, 90.3) | (98.9, 89.5) | (99.3, 94.6) | (97.7, 87.7) | (93.5, 67.1) | (98.1, 93.0) | (97.7, 86.5) | (98.4, 90.9) | (96.5, 91.0) | (97.2, 87.2) |

*Table 10.* Quantitative results on the Real3D-AD dataset (%).

| Object | PointCLIPV2 | | | AnomalyCLIP | | | PointAD | | | CoGeoAD (Ours) | | |
|---|---|---|---|---|---|---|---|---|---|---|---|---|
| | P-AUROC | I-AUROC | I-AP | P-AUROC | I-AUROC | I-AP | P-AUROC | I-AUROC | I-AP | P-AUROC | I-AUROC | I-AP |
| airplane | 50.5 | 45.9 | 46.4 | 50.9 | 48.3 | 51.8 | 71.7 | 62.1 | 68.0 | **84.4** | **73.0** | **74.1** |
| car | 53.6 | 47.0 | 47.2 | 49.6 | 63.3 | 67.7 | 77.8 | 70.1 | 68.6 | **84.6** | **85.6** | **88.3** |
| candybar | 54.2 | 49.4 | 48.7 | 49.8 | 48.8 | 50.1 | 77.0 | 73.8 | 71.7 | **92.4** | **80.9** | **82.7** |
| chicken | 47.0 | 48.3 | 57.1 | 50.1 | 51.9 | 62.0 | 73.2 | 59.4 | 58.5 | **84.6** | **62.9** | **64.6** |
| diamond | 54.1 | 47.6 | 50.1 | 57.5 | 60.2 | 59.2 | 87.6 | **99.6** | **99.6** | **98.2** | 97.2 | 97.0 |
| duck | 60.7 | 64.6 | 62.1 | 47.9 | 42.6 | 46.8 | 60.8 | **69.2** | **67.0** | **95.3** | 60.6 | 61.9 |
| fish | 59.4 | 60.3 | 67.4 | 48.6 | 57.1 | 58.7 | 82.2 | 71.7 | 74.6 | **92.2** | **78.2** | **84.8** |
| gemstone | 51.2 | 60.2 | 60.8 | 48.3 | 54.6 | 58.2 | 85.5 | **87.7** | **86.1** | **97.5** | 81.3 | 74.1 |
| seahorse | 47.1 | 72.4 | 73.8 | 50.2 | 60.0 | 60.1 | 79.4 | 83.1 | 86.8 | **86.0** | **84.0** | **87.7** |
| shell | 55.4 | 60.8 | 58.9 | 50.5 | 51.1 | 50.1 | 79.3 | 92.8 | 93.4 | **97.6** | **95.6** | **96.0** |
| starfish | 41.6 | 60.4 | 58.5 | 49.3 | 40.3 | 45.6 | 79.5 | **90.0** | **93.1** | 79.6 | 83.5 | 86.1 |
| toffees | 47.5 | 72.3 | 74.7 | 51.0 | 53.8 | 58.0 | 80.2 | **86.7** | **89.9** | **95.1** | 76.1 | 83.2 |
| Mean | 51.9 | 57.4 | 58.8 | 50.3 | 52.7 | 55.7 | 77.8 | 78.8 | 79.8 | **90.6** | **79.9** | **81.7** |

