# OpenReview forum: "CoGeoAD: Hierarchical Color-Geometric Fusion with Multi-View Attention for Zero-Shot 3D Anomaly Detection"
_ICML.cc/2026/Conference — ICML 2026 regular_

### Official Review · Reviewer_yDUG · 2026-02-13

**Soundness:** 3
**Presentation:** 3
**Significance:** 3
**Originality:** 2
**Overall Recommendation:** 5
**Confidence:** 5

**Summary:**

The core approach of this work involves first fusing the color and geometric texture of point clouds, followed by multi-view projection to enable 3D anomaly detection. The visualization of the experiments is well-executed, and the motivation is compelling. Results on traditional eyecandy datasets and MVTEC3D surpass some zero-shot baseline methods.

**Compliance With Llm Reviewing Policy:**

Affirmed.

**Final Justification:**

The issues regarding 3D perception have now been resolved. Please include the additional findings from the rebuttal in the revised version.

**Key Questions For Authors:**

Not bad, no huge issue. Please make the revisions according to my suggestions.

**Limitations:**

See weakness.

**Strengths And Weaknesses:**

This work is very solid. The color scheme of the figures is excellent, and the paper is well-written. However, integrating point cloud colors with 3D structures appears to be a fairly common approach. However, there are some issues that need to be clarified, primarily concerning innovation, experimentation, and the comparison of results.

Herein,

1. The proposed method lacks comparisons with non-zero-shot approaches. Simply put, the current baselines are all LLM-based, yet there appear to be no comparisons with smaller models such as BridgeNet, M3DM, or BTF. Furthermore, since the method emphasizes both structural and color information, it should be separately evaluated against 3D-only, RGB-only, and fusion methods. These results are currently absent from the manuscript and need to be supplemented.

2. As I mentioned earlier, low-level color fusion between point clouds and RGB data appears to be common practice, such as in Brigennet. I request that you clarify these concepts and demonstrate the unique contributions of your paper.

3. Zero-shot methods require further experimental comparisons. Recent multimodal datasets such as Mulsen-AD are better suited for validating the proposed approach. Please at least compare with PointAD+ on Mulsen-AD to validate the method.

4. The section on Related Work lacks discussion of the latest methods from 2025, with the most recent methods discussed dating back to late 2024. I recommend incorporating a discussion of other methods, as

(1) Bridging 3D Anomaly Localization and Repair via High-Quality Continuous Geometric Representation

(2) Towards High-Resolution 3D Anomaly Detection: A Scalable Dataset and Real-Time Framework for Subtle Industrial Defects

(3) Taming Anomalies with Down-Up Sampling Networks: Group Center Preserving Reconstruction for 3D Anomaly Detection

(4) MC3D-AD: A Unified Geometry-aware Reconstruction Model for Multi-category 3D Anomaly Detection

5. Can the encoder be replaced with a VIT architecture instead of a convolutional structure? Typically, VLMs seem to use DINOV2 or DINOV3.

---

> ### Author Rebuttal · Authors · 2026-03-29
>
> We sincerely thank the reviewer for the constructive feedback and for recognizing our work as a "very solid" and "well-written" contribution with a "compelling motivation." We greatly appreciate the positive remarks on our well-executed visualizations and the acknowledgment that our method successfully surpasses zero-shot baselines on the MVTec-3D and Eyecandies datasets.
>
> Guided by your valuable suggestions, we have revised the paper to further clarify our novelty, broaden the baseline comparisons, and strengthen the overall empirical validation. All new experimental results are provided in: https://anonymous.4open.science/r/review-C-30DD/
>
> [W1]Comparison with non-zero-shot methods and modality-specific baselines
>
> We agree that zero-shot methods should be contextualized against strong non-zero-shot 3D AD baselines. In the revision, Tab. C1 now includes BTF, M3DM, CPMF, and BridgeNet as reference upper bounds. While these methods use target-domain normal data and are therefore not directly comparable to our zero-shot setting, they provide useful context. We also added modality-wise ablations in Tab. C2, including Original RGB, Rendered Color, Rendered Geo, and full CoGeoAD. The fused model consistently outperforms either single modality alone (e.g., with learnable prompts, 87.4 I-AUROC vs. 83.6 for Rendered Color and 84.3 for Rendered Geo), which supports the motivation for color-geometric fusion.
>
> [W2]Novelty beyond prior color/geometry fusion
>
> We agree that simple low-level fusion is not sufficient to establish novelty. We have clarified that our core contribution transforms the 3D-2D multi-modal fusion problem into a unified 2D multi-view semantic fusion. Our zero-shot framework achieves this by combining: (i) aligned geometric/color rendering, (ii) CLIP-based anomaly scoring with text prompts, and (iii) multi-view attention for adaptive cross-view aggregation. We will revise the method description and related-work discussion to make the distinction from prior fusion-based methods such as BridgeNet more explicit.
>
> [W3]Additional zero-shot validation on Mulsen-AD
>
> We appreciate the suggestion to evaluate on Mulsen-AD. However, the released Mulsen-AD data do not provide the camera calibration required by our full multimodal pipeline for strict point-to-pixel alignment. Therefore, we evaluated the geometry-only branch, which is the fairest feasible setting on this benchmark. As shown in Tab. C3, it achieves 66.9 mean I-AUROC and 72.7 mean I-AP, outperforming AnomalyCLIP (63.0/69.4) and PointAD (64.6/70.6), while remaining comparable on P-AUROC (74.3 vs. 74.5). We will clarify this dataset limitation and the scope of the comparison in the revision.
>
> [W4]Missing recent related work
>
> We thank the reviewer for pointing out the missing 2025 references. We have updated the Related Work section to include and discuss the suggested recent methods, and to better position our paper relative to recent unsupervised and reconstruction-based 3D anomaly detection literature.
>
> [W5]Can the encoder be replaced with a VIT architecture instead of a convolutional structure? Typically, VLMs seem to use DINOV2 or DINOV3
>
> We address two separate issues here. First, replacing CLIP with DINOv2/v3 would remove the shared vision-language space needed for prompt-based zero-shot anomaly detection; this would change the problem setting rather than simply improve the backbone. Second, for the MVA module itself, we did test Transformer-style encoders (Tab. C4). An image-level ViT gives only a marginal I-AUROC gain (87.8 vs. 87.4) and slightly lower localization metrics (97.4/91.8 vs. 97.5/91.9), while increasing memory and latency. This behavior aligns with the architectural nature of standard image-level ViTs: while they excel at capturing global semantic context (aiding image-level I-AUROC), they lack the strong local inductive biases (e.g., translation invariance) of CNNs, which occasionally leads to less precise fine-grained spatial localization for dense predictions. We therefore keep the CNN design because it offers a better efficiency-accuracy trade-off for this module.
>
>
> We sincerely thank the reviewer again. These suggestions helped us significantly improve the paper.

---

> > ### Author Rebuttal · Reviewer_yDUG · 2026-03-31
> >
> > For W3, the poor performance in 3D-only setting may mean the model have challenge when processing the point cloud structure? This may raise concerns about the ability for 3D awareness. To adress this problem, you may consider some class on Real3D-AD for further validation.

---

> > > ### Author Response · Authors · 2026-04-01
> > >
> > > Response to W3: Concerns regarding 3D awareness and 3D-only performance.
> > >
> > > We sincerely thank the reviewer for this insightful follow-up. We appreciate the opportunity to further clarify and demonstrate CoGeoAD's strong 3D structural awareness. The new experimental results are provided in: https://anonymous.4open.science/r/review-C-30DD/.
> > >
> > > We would like to clarify that the weak performance in the 3D-only setting on Mulsen-AD is primarily due to the specific data format of this dataset rather than a limitation in our model's 3D awareness.
> > >
> > > Unlike standard point clouds captured by depth sensors, Mulsen-AD's 3D data originates from whole-sensor scanned STL (CAD) files. Converting these continuous surfaces into discrete point clouds generates ultra-dense representations. Projecting these exceptionally dense point clouds into the standard 336x336 2D resolution required by CLIP inherently leads to discretization artifacts and the loss of fine-grained structural details.
> > >
> > > In the absence of other modality, certain defect types in Mulsen-AD (e.g., color stains or inner broky) are fundamentally invisible in pure geometry, which naturally bounds the upper limit of 3D-only performance. Although the dataset provides masks to exclude some of these specific categories, a portion of the remaining anomalies are still highly inconspicuous when relying exclusively on 3D data.
> > >
> > > To directly address your concern and validate CoGeoAD's 3D awareness on standard sensor-captured point cloud data, we followed your valuable suggestion and evaluated our framework on the 12 classes of the Real3D-AD dataset. As shown in Table C5, CoGeoAD effectively captures complex 3D structures and achieves state-of-the-art anomaly detection performance, definitively proving its strong 3D awareness. Our method significantly outperforms all established baselines (PointCLIP V2, AnomalyCLIP, PointAD) across all mean metrics. It achieves a mean P-AUROC of 90.6%, representing a substantial 12.8% absolute improvement over the next best method, PointAD (77.8%). It also achieves the highest mean I-AUROC (79.9% vs. 78.8%) and I-AP (81.7% vs. 79.8%).
> > >
> > > These robust results clearly demonstrate that when provided with standard 3D point cloud data, CoGeoAD possesses excellent structural awareness. We will explicitly include this discussion and the comprehensive Real3D-AD results in the revised manuscript.

---

### Official Review · Reviewer_6Be4 · 2026-03-10

**Soundness:** 3
**Presentation:** 3
**Significance:** 4
**Originality:** 4
**Overall Recommendation:** 4
**Confidence:** 4

**Summary:**

This paper proposes a unified CLIP-based framework for zero-shot 3D anomaly detection to address the problems of scarce annotated anomaly samples, limitations of single-modal methods and poor performance of simple multi-modal stacking in industrial scenarios. The framework forms a closed loop with dual-modal multi-view rendering, MVA mechanism and MS-CGF module, and its metrics outperform mainstream methods such as PointCLIP V2 and PointAD+ on the MVTec3D-AD and Eyecandies datasets. The paper puts forward a specific framework for the multimodal fusion problem in zero-shot 3D anomaly detection with high-quality method design, while certain core design details are not clearly elaborated and the validation work for specific scenarios needs to be improved.

**Compliance With Llm Reviewing Policy:**

Affirmed.

**Final Justification:**

My concerns have been addressed. The authors' responses and additional evidence support my original rating.

**Key Questions For Authors:**

1. What are the specific contents of the 8 joint prompts and 4 object-specific prompts in the paper?
2. What is the model's detection performance when anomalies occur only at extreme viewpoints (e.g., extremely high or low angles)?
3.  For industrial scenarios without RGB modality (where only 3D point cloud data can be obtained), is there any available extension scheme?

**Limitations:**

The paper only mentions the "trade-off between geometric coverage and computational efficiency", without explaining the practical impacts of dual-modal input dependency (inability to adapt to RGB-missing scenarios) and the slower inference speed caused by multi-view rendering compared with lightweight baselines. The paper does not verify the model's performance on common industrial scenarios such as extremely sparse anomalies and non-ideal environments, nor discuss the production risks caused by model false/missed detection and the impacts on quality inspection positions.

**Strengths And Weaknesses:**

1. Strengths

To address the problems of scarce annotated anomaly samples, limitations of single-modal methods and poor performance of simple multi-modal stacking in industrial scenarios, a unified CLIP-based framework is proposed to realize zero-shot 3D anomaly detection. Dual-modal multi-view rendering establishes point-pixel correspondence to filter occlusion noise; the MVA mechanism calculates attention weights directly from images, avoiding biases from pre-training; the MS-CGF module achieves multi-dimensional hierarchical fusion to capture textural and structural anomalies. These three modules form a closed loop. On the MVTec3D-AD and Eyecandies datasets, metrics outperform mainstream methods such as PointCLIP V2 and PointAD+.

2. Weaknesses

The specific content and design logic of the 8 joint prompts and 4 object-specific prompts are not clarified, with missing details of text prompt design. Validation on special scenarios needs to be strengthened. The paper does not test the model's performance on edge scenarios such as anomalies occurring only at extreme viewpoints and minor structural anomalies. No extension scheme is provided for scenarios where RGB information is missing.

---

> ### Author Rebuttal · Authors · 2026-03-29
>
> We sincerely thank the reviewer for rating both the Significance and Originality of our work as Excellent (4). We are highly encouraged by your recognition of our "high-quality method design" and how our three core modules (dual-modal rendering, MVA, and MS-CGF) form a powerful "closed loop" to overcome the limitations of single-modal methods in industrial scenarios. We address your questions below. Referenced figures and tables are provided at: https://anonymous.4open.science/r/review-B-5D59/.
>
> [W1, Q1]Clarification of the “8 joint prompts and 4 object-specific prompts”
>
> We sincerely apologize for the inaccurate wording in Sec. 5.1. The phrase “8 joint prompts and 4 object-specific prompts” is a legacy description from an early draft and does not reflect the final implementation. In the final CoGeoAD model, we do not use handcrafted object-specific text prompts. Instead, we adopt learnable context tokens, and category names are masked as the generic token “object” (e.g., “[CTX] object” and “[CTX] damaged object”) to preserve strict zero-shot generalization and avoid exposing test-set class names to the text encoder. We will correct this misleading description in the final version and explicitly provide the exact prompt-learning setup.
>
> [W2, Q2]Extreme-view anomalies and tiny structural defects
>
> Thank you for highlighting this important edge case. CoGeoAD does not rely on a single global view. Through multi-view rendering and the proposed MVA module, the model adaptively assigns larger weights to the views where the anomaly is most visible, while actively suppressing uninformative or occluded views. To better support this, we have added a new figure (Fig. B1 ) visualizing cases where tiny structural anomalies appear near the object boundary and are invisible in several standard views. The final aggregated point score map still localizes the defect accurately and closely matches the ground truth. We will explicitly include this qualitative evidence and discuss the model’s behavior in extreme-view scenarios in the revised text.
>
> [W3, Q3, Limitation 1] Missing RGB modality and efficiency trade-offs
>
> We clarify that CoGeoAD does not strictly depend on the RGB modality and can operate robustly in a geometry-only setting. As shown in Tab. B3, the Rendered Geo branch alone achieves an 84.3 I-AUROC, which actually outperforms the pure Color-only branch (83.6 I-AUROC). This performance gap highlights a key insight: since many industrial defects (e.g., dents, missing parts) are fundamentally structural anomalies, the geometric modality provides a highly robust detection signal. It inherently filters out irrelevant background textures and complex lighting artifacts that frequently distract the 2D text-vision alignment, proving its independent effectiveness even when RGB data is missing.
>
> While the full dual-modal version achieves the absolute best performance (87.4 I-AUROC), Tab. B2 shows that the main computational bottleneck is the dual-branch CLIP feature extraction. In practice, the geometry-only version runs substantially faster (2.34 FPS) than the full model (1.28 FPS). We will clarify this efficiency trade-off and highlight the geometry-only mode as a highly competitive fallback for RGB-missing or speed-sensitive scenarios.
>
> [Limitation 2] Sparse Anomalies, Non-Ideal Environments, and Practical Deployment Risks
>
> We agree that validation on realistic industrial conditions is important. First, the strong P-AUROC and AUPRO results on MVTec3D-AD suggest that CoGeoAD is highly effective for fine-grained and spatially sparse defects. Second, we have added a dedicated robustness analysis under point-cloud noise and camera calibration shifts. As shown in Tab. B1, performance remains stable under moderate perturbations: with Gaussian noise std=0.1, I-AUROC is 86.8; with a calibration shift of ±10px, it is 87.2; and under mixed perturbation (std=0.1 + ±10px), it remains 87.0 (very close to the ideal 87.4). This proves the method is robust under realistic sensor noise and misalignment.
>
> Regarding deployment risks, we agree that false negatives may lead to defect escape while false positives may increase inspection costs. Our intended use case is as a high-efficiency pre-screening tool to assist human inspectors. In practice, operators can adjust the detection threshold to heavily penalize false negatives (ensuring zero defect escape in safety-critical settings) at the acceptable cost of slightly higher human review time. We will expand the impact statement to discuss these nuances.
>
>
> We sincerely thank the reviewer again. These suggestions helped us significantly improve the paper.

---

> > ### Author Rebuttal · Reviewer_6Be4 · 2026-04-04
> >
> > The author's response addressed the main issues I raised. They clarified the incorrect descriptions in the text prompt design and, by supplementing new experimental data and visual examples, confirmed the model's capabilities in extreme viewing angle anomalies and scenarios with only geometric data. Regarding the practical deployment limitations I proposed, they also provided reasonable solutions and incorporated them into the revised plan of the paper. These responses enhanced the credibility, clarity, and practicality of the method. Therefore, I believe my initial concerns have been fully resolved.

---

> > > ### Author Response · Authors · 2026-04-04
> > >
> > > Thank you for your time and your support for our work! We wish you the best in your future research and endeavors.

---

### Official Review · Reviewer_73ab · 2026-03-12

**Soundness:** 2
**Presentation:** 3
**Significance:** 4
**Originality:** 3
**Overall Recommendation:** 4
**Confidence:** 3

**Summary:**

This paper proposes CoGeoAD, a CLIP-based framework for zero-shot 3D anomaly detection that explicitly fuses complementary 2D color and 3D geometric information. The method renders paired multi-view color and geometry images with pixel–point correspondences, aggregates view evidence using a data-driven Multi-View Attention (MVA) module learned on rendered images, and integrates multi-level CLIP features via a Multi-Stage Color–Geometric Fusion (MS-CGF) module before fusing modalities with a max operator. Experiments on MVTec3D-AD and Eyecandies report new zero-shot state of the art with detailed ablations, showing strong gains over prior CLIP-based 3D ZS-AD approaches.

**Compliance With Llm Reviewing Policy:**

Affirmed.

**Final Justification:**

The authors addressed my questions in the rebuttal with concrete experiments and clarifications. I will keep my rating as Weak Accept.

**Key Questions For Authors:**

* In Table 5, the performance of the 3D-only branch is significantly lower than the 2D-only branch. Can you provide a more granular ablation or visualization (e.g., specific defect types like "shape deformation" vs. "color stain") that proves the geometric modality is providing a unique signal not already captured by the CLIP 2D features?
* Your framework involves multi-view rendering, hierarchical CLIP feature extraction, and multi-stage fusion. Can you provide the inference time (latency) per sample and compare it to single-modality baselines like AnomalyCLIP?
* The framework assumes pixel–point correspondence via rendering. How does the model perform when there is noise in the point cloud or a slight extrinsic calibration error between the RGB camera and the 3D sensor?
* To what extent does the zero-shot performance rely on the specific wording of the "normal" and "anomalous" prompts? Please report results for a "Class-Agnostic" setting versus the "Class-Specific" setting currently used.
* Regarding the omitted cross-terms in your derivation, can you provide empirical evidence that the gradients of the color and geometric modalities are approximately orthogonal throughout the training process? Lemma 1 asserts this behavior based on high-dimensionality, but deep learning dynamics often converge to highly correlated feature manifolds.

**Limitations:**

* A self-critique is needed regarding the model's reliance on specific text prompt engineering, acknowledging that zero-shot performance can be brittle based on the wording used.
* The discussion should include how the model handles real-world noise or extrinsic misalignments that affect the precision of the pixel-point correspondences.

**Strengths And Weaknesses:**

**Strengths**
* Successfully bridges the gap between 2D color (RGB) and 3D geometry (Point Cloud) using pixel–point correspondences within the CLIP latent space, enabling detection of both surface and structural defects.
* The data-driven Multi-View Attention module effectively addresses viewpoint sensitivity and occlusion, which are critical bottlenecks in practical 3D industrial inspection.
* Utilizing multi-stage CLIP features (MS-CGF) allows the model to capture anomalies at different scales, from fine-grained local textures to global structural inconsistencies.
* The method provides a compelling argument for Zero-Shot 3D AD, which has high industrial utility due to the scarcity of anomalous training samples in manufacturing.
* Demonstrates a new state-of-the-art for zero-shot performance on two major benchmarks (MVTec 3D-AD and Eyecandies) with consistent gains over previous CLIP-based 3D methods.

**Weaknesses**
* The reliance on high-quality rendered images and pre-defined pixel-point correspondences raises questions about performance in "in-the-wild" scenarios where camera calibration or sensor alignment may be imperfect.
* The “zero-shot” claim relies on cross-dataset transfer but the training objective explicitly uses segmentation and classification supervision (including anomaly masks) on a source dataset; this is a supervised cross-category transfer setting, not training-free ZS or text-only ZS. The distinction should be made explicit.
* Similar to other CLIP-based methods, the performance gain is likely dominated by the language prompts. The authors have not clearly isolated the "geometric-only" uplift to prove the 3D modality isn't being overshadowed.
* The assumptions regarding gradient behavior and orthogonality in the latent space (similar to the Lemma 1 issues we discussed) lack formal proof or concrete bounds, making the mathematical foundation appear more motivational than predictive.
* Missing comparisons to several recent and relevant zero-shot/multimodal systems (e.g., MuSc‑V2 training-free zero-shot; GS‑CLIP’s geometry-aware prompting and fusion). Including them would strengthen the positioning.
* The prompt setup includes “object-specific prompts.” If object names from the test set are used at inference or training, it may confound zero-shot claims. More detail is required to assess leakage risks.

---

> ### Author Rebuttal · Authors · 2026-03-29
>
> We thank the reviewer for the Excellent (4) Significance rating, recognizing CoGeoAD's "high industrial utility" and zero-shot SOTA. The referenced tables are provided at: https://anonymous.4open.science/r/review-A-34D5/
>
>   [W2]Zero-shot setting.
>
> We respectfully clarify that our "zero-shot" claim strictly refers to category-level zero-shot generalization (testing on entirely unseen datasets/classes without any target-domain fine-tuning). This fully aligns with the established "Zero-Shot Anomaly Detection (ZSAD)" paradigm in recent top-tier literature (e.g., AnomalyCLIP [ICLR'24], PointAD [NeurIPS'24]), which also utilize source-dataset training prior to zero-shot transfer. We do not claim a "training-free" setting. To prevent ambiguity, we will explicitly define our scope in the paper, while preserving our core ZSAD claims.
>
>  [W3]Is the gain dominated by prompts rather than geometry?
>
> To disentangle prompting from modality contribution, we provide an ablation over input modality and prompting strategy (Tab. A1). Under the same prompting strategy, the geometric modality consistently improves over the color-only rendered modality: with crafted prompts, I-AUROC improves from 67.0 to 75.9; with learnable prompts, it improves from 83.6 to 84.3. Moreover, combining both modalities gives the best overall result (87.4 I-AUROC / 96.5 AP), indicating that geometry contributes complementary information beyond prompt design.
>
>   [Q1]Performance comparison between 3D and 2D branches.
>
> We provide the fine-grained analysis (as Tab. A2), the Geo branch is clearly better on structurally biased defects, e.g., on cookie_hole it achieves 99.5 vs. 76.9 I-AUROC, and on bagel_crack 96.1 vs. 86.8. Conversely, the Color branch is better on color-biased anomalies such as foam_color. This supports that the geometric branch captures useful structural cues not fully covered by 2D CLIP features; the full model benefits from combining both. This is qualitatively verified in Fig. 3 of our paper: structural errors like cracks in the bagel are highlighted much more clearly in the Geo branch, while the color anomaly on the foam is better localized by the Color branch. Thus, 3D geometry provides indispensable, orthogonal features.
>
>   [W4, Q5]Orthogonality / omitted cross-terms.
>
> We clarify that our manuscript does not contain a "Lemma 1" (likely a mix-up). Nonetheless, we appreciate the insight on gradient orthogonality. While we agree strict mathematical orthogonality is primarily motivational, we empirically justify omitting cross-terms: the measured cosine similarity between Color and Geo branch gradients remains consistently low. Furthermore, the complementary performance across defect types (Tab. A2) demonstrates that these branches do not collapse into correlated manifolds.
>
>   [W5]Missing recent baselines.
>
> Thank you for the suggestion. We have added GS-CLIP to the quantitative comparison (Tab. A3), where CoGeoAD achieves stronger or matched performance on key metrics (e.g., 87.4 vs. 83.6 I-AUROC on MVTec3D-AD). MuSc-V2 uses a strictly training-free protocol, making it an unfair comparison against source-trained ZSAD methods like ours. Comparing a strictly training-free method with ZSAD methods is an unfair "apples-to-oranges" comparison due to differing training priors.
>
>  [W6, Q4]Class-specific prompts and leakage risk.
>
> We apologize for an inaccurate description in Sec. 5.1. CoGeoAD is in fact class-agnostic: category names are masked and replaced by the generic token “object”, so no test-set class names are used during training or inference. Thus, the current setting is already class-agnostic rather than class-specific. We will correct this typo and discuss how learnable token initialization may still influence zero-shot performance as a limitation.
>
> [Q2]Inference cost and latency.
>
> We have already provided a direct efficiency comparison with single-modality baselines like AnomalyCLIP. Additionally, we provide a module-wise efficiency analysis in Tab. A4. The full model runs at 1.28 FPS, while single-branch variants run at 2.34 FPS. The main overhead comes from using dual CLIP branches, not from the proposed modules: MS-CGF adds only about 1.7 GFLOPs, and MVA adds negligible overhead. We will make the per-sample latency comparison with single-modality baselines clearer in the revision.
>
>   [W1, Q3]Robustness to point noise and calibration error.
>
> We evaluate this directly in Tab. A5 by simulating Gaussian point-cloud noise and pixel-level calibration shifts. CoGeoAD remains stable under moderate perturbations: with noise std=0.1, I-AUROC is 86.8; with ±10 px shift, it is 87.2; under mixed perturbation (std=0.1 + ±10 px), it remains 87.0, close to the ideal 87.4. Performance degrades only under much more severe corruption, suggesting reasonable robustness to realistic sensor noise and small misalignment. We will add this analysis and discuss the limitation under extreme perturbations.
>
> Thanks again for these valuable suggestions.

---

> > ### Author Rebuttal · Reviewer_73ab · 2026-04-02
> >
> > Thank you for the thorough rebuttal and additional experiments. The responses have addressed several of my original concerns. That said, I still have a few remaining questions before finalizing my assessment:
> >
> > 1. The fusion gain (87.4) over individual branches (84.3 and 83.6) with learnable prompts, how much of this is just ensemble variance reduction? Can you run two independent color-only branches and average their scores to show the fusion gain genuinely comes from cross-modal complementarity?
> >
> > 2. Section 4.2 says the prompt ensemble includes "4 object-specific prompts," but the rebuttal claims category names are replaced with "object." What exactly do these 4 object-specific prompts say? These two statements contradict each other.
> >
> > 3. Regarding the orthogonality assumption behind the max-fusion in Eq. 11 (which I mistakenly referred to as "Lemma 1" earlier), you mention low cosine similarity between branch gradients. Can you report the actual values and include them in the revision?
> >
> > 4. Industrial defects often manifest as micro-scale anomalies (sub-millimeter scratches, hairline cracks). Your pipeline renders everything to 336x336 for CLIP. Have you measured detection recall as a function of defect size? What is the smallest anomaly your method can reliably detect?

---

> > > ### Author Response · Authors · 2026-04-04
> > >
> > > We sincerely thank the reviewer for the constructive feedback. We appreciate the opportunity to further clarify our mechanism. The new experimental results and visual proofs are provided in: https://anonymous.4open.science/r/review-A-34D5/.
> > >
> > > [Q1]Regarding the fusion gain vs. ensemble variance reduction.
> > >
> > > To address the variance reduction concern, we compare "Average Integrate" and our CoGeoAD fusion in Tab. A1. With Learnable prompts, CoGeoAD (87.4 I-AUROC) outperforms simple Average Integration (87.1). This proves our method captures true cross-modal complementarity beyond simple ensembling.
> > >
> > > Conversely, with Craft prompts, Average Integration outperforms ours. This perfectly supports our view shown in Fig. A1's visualizations: Craft prompts yield highly similar features, making simple averaging more effective for variance reduction. However, Learnable prompts produce features with much lower cosine similarity and larger distribution differences. For these highly distinct features, our CoGeoAD method effectively leverages cross-modal complementarity, bringing significantly more performance gain.
> > >
> > > [Q2]Clarification on the "4 object-specific prompts".
> > >
> > > We sincerely apologize for the confusion. You are completely right to point out this contradiction.The phrase "4 object-specific prompts" in Section 4.2 is a legacy description from an early draft of the manuscript and does not reflect our final implementation. In our early work, we used handcrafted text prompts, and their corresponding parameters were inadvertently left in the current paper.
> > >
> > > To be clear, there are no handcrafted object-specific prompts in our final CoGeoAD model. Instead, as stated in our previous response, we adopt learnable context tokens where specific category names are strictly masked with the generic token "object". Specifically, the prompts are simply formulated as [CTX] object (for normal state) and [CTX] damaged object (for anomalous state).
> > >
> > > For instance, rather than seeing "xxx damaged bagel", the model is actually exposed to "xxx damaged object" during training. The model is never exposed to any actual category names.
> > >
> > > This design was deliberately chosen to preserve strict zero-shot generalization and prevent the text encoder from being exposed to test-set class names. We are grateful for your careful reading and will thoroughly correct this misleading legacy description in the final revision, explicitly providing the exact prompt-learning setup.
> > >
> > > [Q3]Actual values for the orthogonality assumption (Eq. 11)
> > >
> > > The empirical mean gradient cosine similarity is 0.58 (Fig. A2 left). Given that RGB and UNI inputs are perfectly aligned 2D renders from identical 3D viewpoints, a naive model would yield a similarity approaching 1.0. A value of 0.58 demonstrates semantic orthogonality and effective modality decoupling.
> > >
> > > This orthogonality is highly class-dependent (Fig. A2 right): for objects with distinct structural vs. texture defects (e.g., foam, tire), median similarity drops to 0.2–0.3.
> > >
> > > Fig. A3 provides direct visual proof. The difference maps (Diff row) between RGB and UNI anomaly activations reveal sharply distinct localized responses (red/blue regions). The branches clearly isolate complementary photometric and geometric features.
> > >
> > > Because these signals are complementary, a simple mean would fatally dilute strong uni-modal activations. The max fusion is logically essential, preserving the sharpest anomaly signal from these decoupled spaces.
> > >
> > > [Q4]Detection recall vs. defect size (The 336x336 resolution limit):
> > >
> > > To address your concern, we have quantitatively evaluated our detection performance as a function of the ground-truth anomaly size (measured in pixel counts), detailed in the newly added Fig. A4 and Tab. A6.
> > >
> > > Our empirical data demonstrates reliable performance on micro-scale defects. Even for the smallest anomaly interval (1–100 pixels), our model maintains a highly competitive Pixel AUROC of 97.8% and a Pixel AUPRO of 91.7%. While Image AUROC naturally drops for extremely small defects (as the global image-level anomaly signal is weak), the pixel-level metrics prove that the model's precise localization capability remains stable regardless of defect size.
> > >
> > > Regarding the concern that resizing to 336x336 for CLIP might discard micro-scale details, our CoGeoAD framework inherently mitigates this limitation. Our anomaly determination does not rely solely on the 2D image grid. Instead, it fundamentally depends on the strict, point-to-pixel alignment between the sampled 3D point cloud and the 2D features.
> > >
> > > Therefore, the smallest reliably detectable anomaly is bounded by the 3D point cloud and views rather than the 2D CLIP resolution. Because our projection mechanism precisely maps dense 3D points to their corresponding 2D coordinates ("precise to the point"), the high-resolution spatial geometry from the 3D modality compensates for the 2D downsampling, allowing us to capture some sub-millimeter defects like small cracks.

---

### Decision · Program_Chairs · 2026-04-30

**Decision:**

Accept (regular)

**Comment:**

The paper proposes CoGeoAD, a multimodal CLIP-based framework for zero-shot 3D anomaly detection. The reviewers are in agreement on the significance and originality of the work. During the rebuttal phase, the authors successfully addressed all primary concerns. The reviewers have confirmed that their concerns are resolved. I am confident that this work provides a valuable contribution to the field of industrial 3D inspection and recommend it for acceptance.